# A global climatology of the ocean surface during the Last Glacial Maximum mapped on a regular grid (GLOMAP)

André Paul[1], Stefan Mulitza[1], Ruediger Stein[1,2], and Martin Werner[2]

[1] MARUM – Center for Marine Environmental Sciences and Department of Geosciences, University of Bremen, Bremen, Germany

[2]Alfred Wegener Institute, Helmholtz Centre for Polar and Marine Research (AWI), Bremerhaven, Germany

**Correspondence:** André Paul (apaul@marum.de)

**Abstract.** We present a climatology of the near-sea surface temperature (NSST) anomaly and the sea-ice extent during the Last Glacial Maximum (LGM, 23,000–19,000 years before present) mapped on a global regular $1° \times 1°$ grid. It is an extension of the Glacial Atlantic Ocean Mapping (GLAMAP) reconstruction of the Atlantic NSST based on the faunal and floral assemblage data of the Multiproxy Approach for the Reconstruction of the Glacial Ocean Surface (MARGO) project and several recent estimates of the LGM sea-ice extent. Such a gridded climatology is highly useful for the visualization of the LGM climate, calculation of global and regional NSST averages and estimation of the equilibrium climate sensitivity, as well as a boundary condition for atmospheric general circulation models. The gridding of the sparse NSST reconstruction was done in an optimal way using the Data-Interpolating Variational Analysis (DIVA) software, which takes into account the uncertainty on the reconstruction and includes the calculation of an error field. The resulting Glacial Ocean Map (GLOMAP) confirms the previous findings by the MARGO project regarding longitudinal and meridional NSST differences that were greater than today in all oceans. Taken at face value, the estimated global and tropical cooling would imply an equilibrium climate sensitivity at the lower end of the currently accepted range. However, because of anticipated changes in the seasonality and thermal structure of the upper ocean during the LGM as well as uneven spatial sampling the estimated cooling and implied climate sensitivity are likely to be biased towards lower values.

## 1 Introduction

Gridded climatologies are useful for a number of purposes, for example, for visualizing present or past climate states, calculating global and regional averages, or evaluating climate models. Regarding the evaluation of climate models, unless data locations and model grid points coincide, we cannot quantify the data-model misfit without any sort of mapping. Thus, sparse data must be mapped onto the model grid by statistical methods (Schäfer-Neth et al., 2005; Marchal and Curry, 2008). Furthermore, a gridded sea-surface temperature (SST) climatology may serve as a boundary condition for atmospheric general circulation models (AGCMs) and enable a model evaluation that does not depend on the quality of a simulated SST climatol-

ogy, allowing for another approach in comparing coupled climate models such as in the Paleo-Model Intercomparison Project (PMIP, e.g., Kageyama et al., 2017; Kageyama et al., 2021).

A climate state of the past that is particularly useful for evaluating climate models is the Last Glacial Maximum (LGM, 19,000 to 23,000 years before present; Mix et al., 2001) cold period: The radiative perturbations due to changes in insolation, greenhouse gases and ice sheets are relatively well defined and the paleo-data coverage is comparitively dense and indicates a large response to the radiative forcing (Jansen et al., 2007; Masson-Delmotte et al., 2013).

Previous work on a gridded near-sea surface temperature (NSST) climatology for the LGM includes the Climate: Long

range Investigation, Mapping, and Prediction (CLIMAP) project (CLIMAP Project Members, 1981), the Glacial Atlantic Ocean Mapping (GLAMAP) reconstruction of the Atlantic NSST (Sarnthein et al., 2003a) and the Multiproxy Approach for the Reconstruction of the Glacial Ocean Surface (MARGO, Kucera et al., 2005a). While CLIMAP and GLAMAP (Paul and Schäfer-Neth, 2003; Schäfer-Neth and Paul, 2004) provide seasonal reconstructions of the Earth's surface at the LGM mapped on a 2° grid, MARGO only performed a "pseudo gridding" by calculating 5° block averages (MARGO Project Members,

2009).

Following, e.g., Dail and Wunsch (2014), the adjective "near" is used to distinguish these temperatures, which in the case of the GLAMAP and MARGO projects are based on calibrations for the top 10 m of the ocean and depend on phytoplankton and zooplankton that live even deeper, in the top 200 m to 300 m of the ocean, from those used in other communities, in which the SST is at the surface itself and can even be a skin temperature that does not reflect the temperature below.

CLIMAP used a subjective analysis procedure (i.e. contouring by hand) to yield the paleoisotherm maps (CLIMAP Project Members, 1976, 1981), which were then digitized on a regular grid (Broccoli and Marciniak, 1996; Manabe and Broccoli, 2020). With respect to GLAMAP, different methods were applied: Contouring of the paleotemperature maps was also by hand, and the isotherms were derived by means of visual triangulation from strictly linear interpolation between the NSST reconstructions at the irregularly distributed neighbor sites (Sarnthein et al., 2003a; Pflaumann et al., 2003). For gridding, either

the digitized isotherms (Paul and Schäfer-Neth, 2003) or the NSST reconstructions at the sediment core positions (Schäfer-Neth and Paul, 2004) were objectively interpolated using variogram analysis and kriging in spherical coordinates; and the resulting gridded fields were compared (Schäfer-Neth and Paul, 2004, Fig. 5). The seasonal cycle was constructed following the PMIP (1993) guidelines: A sinusoidal cycle was fitted to the glacial-to-modern anomalies and then the modern monthly NSST (taken as 10 m data from the WOA, 1998) was added. The variogram analysis and kriging cannot deal easily with coastlines, for

example, it may take into account data points separated by a land bridge or an island. This was one motivation to apply the DIVA method (Troupin et al., 2012), which employs a finite-element mesh derived from a given topography.

Here we present an ocean climatology of the sea surface during the LGM mapped on a global regular $1° \times 1°$ grid. This Glacial Ocean Map (GLOMAP) extends the gridded GLAMAP climatology to the global ocean based on the MARGO NSST reconstruction. In addition, we included a more recent estimate of Southern Ocean summer sea-ice extent (Roche et al., 2012)

and reconstructions of Arctic and North Pacific sea-ice extent using the IP25/PIP25 sea-ice proxy and phytoplankton-derived biomarkers (Xiao et al., 2015; Méheust et al., 2016; Méheust et al., 2018). The sparse NSST reconstruction, complemented with the reconstructed sea-ice boundaries in the northern and southern hemispheres, was gridded in an optimal way using the

Data-Interpolating Variational Analysis (DIVA) method (Troupin et al., 2012). This method allows one to take into account the uncertainty on the (paleo) data and calculate an uncertainty field, which can be used as a weight in calculating uncertainty-weighted global and regional averages. Originally developed for usually much denser oceanographic observations, the DIVA method proved to be capable of analyzing sparse paleo data as well.

## 2 Methods

### 2.1 Selecting LGM sea-ice extent reconstructions

For estimating LGM sea-ice extent, we made use of estimates of maximum and minimum sea-ice extent in the Northern and Southern Hemispheres and added a physically reasonable seasonal cycle. As for the MARGO reconstruction of sea-ice extent in the northern North Atlantic Ocean, all four proxies used (planktonic foraminifer assemblages, dinocyst assemblages, alkenone coccolithophorid biomarkers and Mg/Ca ratios in planktonic forminifers) support the same features of sea-ice cover (de Vernal et al., 2006; cf. Sarnthein et al., 2003b). The IP25/PIP25 data by Xiao et al. (2015, Fig. 7a) and Méheust et al. (2018) add information for the Barents Sea and the North Pacific Ocean, respectively. Since there are only few NSST reconstructions in the high latitudes of either hemisphere, the information on past sea-ice coverage also served to fill in the gaps.

In line with earlier transfer function results obtained by the GLAMAP project (Sarnthein et al., 2003a), the Nordic Seas were taken as ice-free during summer. Since the sea-ice edges are not provided in digital format, we digitized the curves from the published maps by de Vernal et al. (2005, Fig. 10, upper left) and Kucera et al. (2005b, Fig. 25) for the Labrador Sea and Nordic Seas, Xiao et al. (2015, Fig. 7a) for the Barents Sea and Méheust et al. (2018, Fig. 9a) for the North Pacific Ocean to obtain their location in geographic coordinates. In case of Xiao et al. (2015, Fig. 7a), neither the projection nor the coordinates are given, hence we used the few indicated topographic features (islands) and the sediment core locations to take into account the summer ice edge north of the Barents Sea in our sea-ice mask. Similarly, we digitized sea-ice reconstructions by Gersonde et al. (2005, Fig. 4, maximum extent of winter sea ice "E-LGM-WSI" and sporadic occurrence of summer sea ice "E-LGM-SSI") and Roche et al. (2012, Fig. 4, bottom, Southern Ocean summer sea-ice extent "PROX.") for the Southern Hemisphere. If necessary, we re-projected them (e.g., from a polar stereographic or orthographic projection) to longitude and latitude. We connected them smoothly in each hemisphere and season and created sea-ice masks for summer and winter (note that the sea-ice reconstructions based on IP25/PIP25 by Xiao et al., 2015, and Méheust et al., 2018, apply to spring, but were in this study taken as an approximation to the winter sea-ice edge).

We created a seasonal cycle of sea-ice coverage as follows: At a given longitude, we assumed a sinusoidal cycle of the latitude of the sea ice edge (cf. Eisenman, 2010) between the maximum and minimum sea ice extent in either hemisphere.

### 2.2 Selecting LGM NSST reconstructions

Faunal and floral assemblages still provide the best spatial coverage and they are the only sedimentary proxy that has the potential to provide a seasonal reconstruction. For these reasons, as well as for internal consistency and reduced noise among

the individual LGM estimates, we selected from the MARGO database (Kucera et al., 2005a; MARGO Project Members, 2009) only those that were based on the faunal and floral transfer function technique. However, in the Nordic Seas, there are large discrepancies between the different NSST reconstructions, well above their level of uncertainties (de Vernal et al., 2006). We therefore used dinocyst assemblages only south of the assumed winter sea-ice boundary at about 50°N (de Vernal et al., 2005; de Vernal et al., 2006). To this end, we extracted all dinoflagellate data assumed to be not affected by winter sea-ice cover in the Nordic Seas.

Each MARGO NSST estimate is associated with an error that is equal to the product of the calibration error and a semi-quantitative "reliability index". The reliability index takes into account the number of samples, the quality of the age model and a possible lack of stationarity reflecting, for example, possible no-analogue situations and a known regional or sedimentological bias (MARGO Project Members, 2009). The calibration error ranges typically between 1 °C and 1.5 °C, and the reliability index ranges between 1 for high reliability and about 3.3 for low reliability. All errors were taken to reflect a $1\sigma$ confidence interval.

## 2.3 Gridding

We chose DIVA over other methods because it takes the coastlines into account, since the analysis is carried out on a finite-element mesh that is restricted to the sea. This prevents the exchange of information across boundaries such as land bridges, peninsulas or islands, which otherwise might produce artificial mixing between, for example, Pacific and Atlantic water masses across the Panama isthmus. We used version 4.7.1 (doi:10.5281/zenodo.836727) of DIVA (Troupin et al., 2012). The purpose of DIVA is to satisfy a variational principle that includes the magnitude of the data (anomalies) themselves as well as the gradients, the spatial variability and data-analysis misfits (Troupin et al., 2019, Eqs. 2.10 and 2.11). Thus in solving the variational principle, DIVA not only takes into account the distance between analysis and data, but also imposes a smoothness constraint and, if desired, an advection constraint. Moreover, it provides an uncertainty estimate.

The general work flow of DIVA is summarized in Figure 3. The first step is to generate coastlines from a given topography. Based on the resulting coastlines, a finite-element mesh is created, which in our application covers the global ocean including the sea ice. Then first-guess values of the three analysis parameters (correlation length, signal-to-noise ratio and variance of background field) are estimated and an analytic covariance function is fit to the data, yielding a revised estimate of the correlation length. ~~Finally, a~~A generalized cross validation can be carried out to improve the the estimates of the signal-to-noise ratio and the variance of background field. Finally, the analysis itself is performed, using the estimated parameters.

To first test the DIVA method on data that are much sparser than oceanographic observations, we adopted the procedure by Schäfer-Neth et al. (2005). We took the test data from the World Ocean Atlas 1998. According to WOA (1998), the original ocean profile data are first vertically interpolated from observed depth levels to standard depth levels, then the arithmetic means of each variable in each 1° and 5° square of the World Ocean are calculated. Except for calculating the arithmetic means, these data have not been subject to any other analysis. These global fields are therefore referred to as "unanalyzed" fields. The 1° unanalyzed annual-mean temperature at a depth of 10 m had been used to calibrate the MARGO transfer function technique (the original data file name is `t00mn1`; it is also available as `otemp.raw1deg.nc` from psl.noaa.gov).

Schäfer-Neth et al. (2005) further binned these data into a regular grid with a constant resolution of 2° using the GMT program xyz2grd (Wessel and Smith, 1998). The coverage is nearly complete, except for the central Arctic Ocean and some points off the Antarctic coast (cf. Fig. 1, *top*). Finally, they greatly reduced this coverage by keeping only those 2° squares that contain an ocean sediment core site from MARGO Project Members (2009). This is the input data set for testing the DIVA method (cf. Fig. 1, *middle*).

The DIVA method was used to interpolate these sparse test data to a complete regular grid with a constant resolution of 2°. The differences between the interpolated field (Fig. 1, *bottom*) and the "unanalyzed" field (Fig. 1, *top*) were calculated as a measure of the misfit. This allows for a near-global assessment of the result from the interpolation using the DIVA method.

To apply the DIVA method to the paleo data, we used the glacial topography GLAC-1D at 21,000 years before present (cf. Tarasov et al., 2012; Briggs et al., 2014) to generate glacial coastlines and create a corresponding global finite-element mesh using a cosine projection. The first-guess values of the correlation length and the signal-to-noise ratio were set to 10° and 1.0, respectively. The first-guess value of the variance of the background field was estimated from the foraminiferal NSST reconstructions as 6.3 $(°C)^2$. We fitted the covariance function to the 444 foraminiferal data points for the nominal seasons January-February-March (JFM) and July-August-September (JAS) and obtained estimates of the correlation length of 9.2° and 10.2°, respectively. The data covariance for JAS was overall larger than for JFM, resulting in a slightly larger correlation length. In the remainder of our study, we fixed the correlation length at an average value of 10°. The generalized cross validation did not yield significantly different values for the signal-to-noise ratio and the variance of background field. To each data value, we assigned a relative weight, which was inversely proportional to the error (a large value corresponded to a high confidence) and normalized such that the sum over all inverse relative weights equaled the number of data points.

We performed two iterations to create a global gridded climatology of monthly NSST. Two iterations were necessary in order to make use of the diatom and radiolarian data from the Southern Ocean, which were only available for Southern Hemisphere summer (JFM), because in this region the biogenic particle flux to the sea floor is restricted to austral summer, even in areas unaffected by sea-ice cover (Abelmann and Gersonde, 1991; Gersonde and Zielinski, 2000; Fischer et al., 2002). Therefore, in the first step, we only used the foraminiferal and dinoflagellate data for JAS and JFM (464 data points). In the second step, we included the diatom data (117 data points) and radiolarian data (19 data points) available for JFM and filled in the missing data for JAS by taking the results from the first step at the grid points where diatom and radiolaria data for JFM exist. In this way we were able to create monthly data at all grid points where data exist and repeat the DIVA analysis:

1. In the first iteration, we concatenated all foraminiferal data and the dinoflagellate data for ice-free regions including their relative weights for JAS and JFM. We created seasonal (monthly) data from the JFM (taken as February) and JAS (taken as August) data using a sine function. We extracted the geographic positions marked as sea ice from the monthly masks of the reconstructed sea-ice extent and determined the local NSST anomaly using a temperature of -1.8 °C for the LGM value and the World Ocean Atlas (WOA, 1998) at 10 m depth for the modern value. To each sea-ice covered data point we assigned an error of 2 °C that was chosen to be larger than the error of any individual LGM estimate from the MARGO database to reflect the uncertainty in the LGM sea-ice extent reconstructions. Then we concatenated the seasonal (monthly) foraminiferal and dinoflagellate data and the local NSST anomalies from the sea-ice reconstructions

and normalized the individual errors such that the sum of all errors equaled one [all NSST anomalies are relative to WOA (1998), which was used by the MARGO Project Members (2009) for calibrating the methods for estimating the LGM NSST values]. Finally, we gridded the data for each month. We achieved continuity across the 0° meridian by adapting the method by Tyberghein et al. (2012) and running two DIVA analyses, one ranging from 0° to 360° in longitude (on the "original grid"), the other ranging from -180° to 180° in longitude (on a "shifted grid"). The two resulting analyses were combined in one on an output grid that extended from 0° to 360° in longitude by calculating a weighted average for each grid point, where the weights were proportional to the zonal distance from the central longitude of the respective input grid.

2. In the second iteration, new (artificial) diatom and radiolarian data for Southern Hemisphere winter (JAS) were generated at the grid points where diatom and radiolarian data for Southern Hemisphere summer (JFM) exist, either using the anomaly with respect to the present observed NSST or the gridded data from iteration 1, depending on whether a grid point was assumed to be ice-covered or ice-free for the LGM. We again created seasonal (monthly) data from the February (JFM) and August (JAS) data using a sine function and concatenated all the seasonal (monthly) data as before, but now including the diatom and radiolaria data, and we carried out two more DIVA analyses, one for the original and one for the shifted grid. Finally, the two grids were merged once more following the method of Tyberghein et al. (2012).

## 2.4 Comparison to other reconstructions

For a comparison to other reconstructions, we selected the recent studies by Annan and Hargreaves (2013), Kurahashi-Nakamura et al. (2017) and Tierney et al. (2020) as well as the earlier studies by CLIMAP (1981) and GLAMAP (Sarnthein et al., 2003a). The horizontal resolution differs among these reconstructions and ranges between 1° and 5°. For analysis and plotting purposes, we interpolated them to the same regular grid with a constant resolution of 1°. We calculated the annual-mean anomalies for the global, tropical and high-latitude oceans from these studies as well as our own results. Because an uncertainty estimate was not available for all studies, we only weighted by area.

## 3 Results

### 3.1 Test of the DIVA method

Figure A1 shows the coastlines that were generated from the modern topography (based on the bottom depth assigned to each 1° square by Garcia et al., 2019) for testing the DIVA method on the WOA (1998) data sampled at the MARGO core locations, as well as from the LGM topography (GLAC-1D, cf. Tarasov et al., 2012; Briggs et al., 2014) for our application of the DIVA method to the LGM NSST reconstruction. The figure also shows the finite-element meshes based on these coastlines, which exhibit a rather homogeneous resolution.

In Table 1, our results of testing DIVA on a sparse and irregularly spaced subset of WOA (1998) are compared in terms of the root-mean square differences for the different oceans as well as the global ocean by Schäfer-Neth et al. (2005) to two

other interpolation methods: variogram analysis and kriging (Deutsch and Journel, 1992) and a variant of objective analysis ("iterative difference-correction method", Levitus, 1982). Furthermore, Fig. 2 shows a map of the absolute difference that can be directly compared to Fig. 5 by Schäfer-Neth et al. (2005). The regionally averaged results from DIVA turned out to be very similar to those from variogram analysis and kriging and better than those from the Levitus objective analysis (which depends strongly on the data coverage and the quality of the zonal-mean that serves as the first guess). The comparison of Fig. 2 and Fig. 5 by Schäfer-Neth et al. (2005) supports this finding at the local scale: Similar to the variogram and kriging results (Schäfer-Neth et al., 2005, Fig. 5 top), absolute differences were generally small in regions of dense spatial sampling. They became particularly large in the Gulf Stream and Kuroshio regions, because of coarse spatial sampling and (probably) the impact of advection by the western boundary currents, which is missing in our application of DIVA.

## 3.2   Patterns of LGM NSST change

The monthly maps based on the gridded MARGO NSST anomaly clearly exhibit the same basic patterns of LGM NSST change as the original MARGO Project Members (2009) synthesis (Fig. 4 and Figs. A2 to A9): Generally, the cooling was larger in the Atlantic than in the Pacific and Indian Oceans. There were strong longitudinal and latitudinal differences in all oceans. The cooling was generally larger in the eastern parts of the oceans than in the western parts and was particularly expressed along the coast of Africa, possibly due to an increase in upwelling or an eastward shift of the coastline and the coastal upwelling systems off Northwest and Southwest Africa (cf. Giraud and Paul, 2010). There was even a 1 °C to 3 °C cooling in the western Pacific warm pool, but overall the east-west temperature differences were less pronounced in the tropical Pacific and Indian Oceans than in the tropical Atlantic Ocean. A 2 °C to 6 °C cooling in the Southern Ocean may indicate a northward migration of the Polar Front. The apparent warming by 1 °C to 2 °C of the subtropical gyres in the Pacific Ocean was associated with a rather large uncertainty.

## 3.3   Global and regional mean changes

Annual averages of the gridded monthly values and their uncertainties were calculated as arithmetic means without any weighting. Global and regional averages (Table 2) and zonal averages (Fig. 5) were calculated from the annually-averaged values as weighted means

$$\overline{T} = \frac{\sum_i w_i T_i}{\sum_i w_i},$$ (1)

where the weights were given by

$$w_i = \frac{A_i}{u_i^2}$$ (2)

with $A_i$ the area of the $i$th grid cell and $u_i$ the uncertainty of the gridded value in the $i$th grid cell. The uncertainties of the global and regional averages were estimated as

$$\overline{u} = \sqrt{\sum_i \left( \frac{w_i}{\sum_i w_i} u_i \right)^2 \times f_{\text{inf}}}\,.$$ (3)

Here the first factor is the simple sum of the local values of the uncertainty of the gridded field that neglects any spatial covariances (i.e., the non-diagonal terms of the covariance matrix), and the second factor is applied to take into account the missing spatial covariances in an approximate way. According to Troupin et al. (2019), the inflation factor

$$f_{\text{inf}} = \sqrt{\frac{4\pi L^2}{\Delta x \Delta y}} \tag{4}$$

is probably too high, yielding overestimates of the uncertainties. With $L = 10°$ the correlation length of the analysis and $\Delta x = \Delta y = 1°$ the resolution of the grid, in our case the numerical value of the inflation factor was $f_{\text{inf}} = 35.45$.

According to Table 2, the global LGM decrease in the gridded NSST was (1.7±0.1) °C. The global tropics (taken to be between 30° S and 30° N) cooled on average by (1.2±0.3) °C, but the tropical Atlantic Ocean by about (1.8±0.6) °C. The cooling in the mid- to high-latitudes was around (3.1±0.2) °C in the North Atlantic Ocean and around (1.4±0.3) °C in the South Atlantic Ocean.

### 3.4 Changes in the meridional differences

The change in the tropical meridional NSST difference was calculated as the average NSST anomaly between the equator and 30° N minus the average NSST anomaly between 30° S and the equator (cf. McGee et al., 2014). According to Table 2 and standard uncertainty propagation, this difference decreased by (0.4±0.6) °C for the global ocean and by (0.4±1.2) °C for the Atlantic Ocean.

In contrast, the meridional NSST difference between the mid- to high-latitudes in the North Atlantic Ocean (north of 45° N) and the South Atlantic Ocean (south of 30° S - these two regions were chosen in accordance with Rahmstorf, 1996) increased by (1.7±0.3) °C.

### 3.5 Zonal-mean changes

The zonal mean changes for the global ocean and the Atlantic, Pacific and Indian oceans are shown in Fig. 5. Changes in the Atlantic Ocean were larger than in the other oceans, and changes in the mid- to high latitudes were larger than in the low latitudes, except for the tropical South Atlantic Ocean, where they reached -4 °C due to the cooling in the coastal and equatorial upwelling regions.

### 3.6 Data-analysis misfit

The normalized data-analysis misfit was determined as

$$J_{\text{misfit}} = \frac{1}{N_{\text{data}}} \sum_{i=1}^{N_{\text{data}}} \frac{\left(T_i^{\text{gridded}} - T_i^{\text{data}}\right)^2}{e_i^2} , \tag{5}$$

where $N_{\text{data}}$ is the number of data-analysis pairs and $e_i$ is the average uncertainty of the data in the $i$th grid cell. The normalized misfit was $J_{\text{misfit}} = 1.5$ with $N_{\text{data}} = 420$ for JAS and $J_{\text{misfit}} = 1.7$ with $N_{\text{data}} = 528$ for JFM, respectively. The geographic

distribution of the individual misfits at the data locations is shown in Fig. 6. Values larger than the uncertainty of the original data occur in the coastal and equatorial upwelling regions and near and under the reconstructed sea-ice cover.

## 3.7 Comparison to other reconstructions

Table 3 lists the recent studies by Annan and Hargreaves (2013), Kurahashi-Nakamura et al. (2017) and Tierney et al. (2020) and compares them to our study in terms of the data, model(s) and method used. Maps of annual-mean sea-surface temperature anomalies are shown in Fig. 7. According to Table 4, the global ocean cooling across the different reconstructions ranged from 1.5 °C to 3.6 °C, while tropical ocean cooling ranged from 0.9 °C to 3.4 °C. For the Atlantic Ocean with better data coverage than the Pacific Ocean, the tropical cooling was between 1.6 °C and 3.7 °C. In all three cases, the lowest value was from CLIMAP Project Members (1981) and the highest from Tierney et al. (2020). All reconstructions show an amplified cooling in the Atlantic Ocean north of 45° N, with the maximum cooling of 7.1 °C given by CLIMAP (1981).

## 4 Discussion

The main purpose of our study was to demonstrate the applicability of the DIVA method to sparse paleo data and provide a gridded NSST reconstuction for the testing of coupled climate models and forcing of AGCMs. We indeed found that the DIVA method was capable of analyzing data that were much sparser than current oceanographic observations, with a skill that was comparable to variogram analysis and kriging, but thanks to the underlying global finite-element mesh with less complications (such as the introduction of communication masks to avoid the pairing of data points that are unlikely to influence each other in the real ocean, cf. Schäfer-Neth et al., 2005, Fig. 2) and overall in less time. Figures 4 and 6 show that when applied to the paleo data the interpolated fields are neither "noisy" nor "patchy". Because the paleo data allowed for a large correlation length of 10°, we obtained a smooth climatology, which we take as an indication that the data points were not overfitted. In addition, our gridded data set of LGM NSST anomalies allowed us to evaluate changes in global and regional averages and spatial differences including their uncertainties.

Following Eq. 3, we calculated the uncertainties of the global and regional averages as the product of the simple sum of the diagonal terms of the error covariance matrix and an inflation factor, which probably resulted in overestimates. In fact, DIVA may be used to more accurately estimate the spatial covariances as described by the non-diagonal terms, albeit at a much higher computational cost (Troupin et al., 2012; Beckers et al., 2014; Troupin et al., 2019, Section 4.5; see also Wunsch, 2018). Therefore we decided to use the simplified inflation approach. We obtained a mean change for the global ocean of (-1.7±0.1) °C and a mean change for the tropical ocean between 30° S and 30° N of (-1.2±0.3) °C. As compared to MARGO Project Members (2009, Table 1), these values tend to be smaller by 0.2 °C to 0.3 °C, possibly because the MARGO results are based on block-averaged NSST anomalies with an incomplete coverage biased towards the eastern continental margin, while our results are based on complete fields obtained from the DIVA analysis.

Our result of a change in the tropical meridional NSST difference by (0.41±0.6) °C reflects a greater cooling in the southern tropics than in the northern tropics, mainly due to changes in the coastal and equatorial upwelling regions. It is consistent with

the original MARGO synthesis (cf. MARGO Project Members, 2009, Figs. 3 and 4), but inconsistent with, e.g., McGee et al. (2014) who estimate a change of (-0.14±0.18) °C that indicates a greater cooling in the northern tropics than in the southern tropics. According to McGee et al. (2014, Fig. 3), our result would correspond to a northward shift of the ITCZ by (0.8±1.3)° and a decrease of the cross-equatorial heat transport by (0.31±0.5) PW, while McGee et al. (2014) obtain a southward shift by (-0.29±0.38)° and an increase by (0.11±0.14) PW. Part of the differences may be due to our denser data coverage and that we based our calculation of regional averages on a gridded analysis as opposed to single ocean sediment cores or block averages with incomplete coverage. However, we stress that strictly speaking neither our results nor the results by McGee et al. (2014) are statistically significant, because the inferred changes are smaller than their estimated uncertainties.

In contrast, the increase of the meridional NSST difference between the mid- to high-latitudes in the North Atlantic Ocean and the South Atlantic Ocean of (1.7±0.3) °C is statistically significant and by itself would argue for an intensified Atlantic meridional overturning circulation, which is indeed found in some simulations of the LGM ocean circulation (e.g., Kurahashi-Nakamura et al., 2017). However, this increase may be counteracted by an accompanying decrease of the sea-surface salinity gradient, which may result in an overall decrease of the sea-surface density gradient (e.g., Paul and Schäfer-Neth, 2003). Both the decrease of the tropical meridional NSST difference and the increase of the large-scale Atlantic meridional NSST difference are also evident from the zonal-mean NSST changes in Fig. 5.

The normalized misfits of $J_{\mathrm{misfit}} = 1.5$ for JAS and $J_{\mathrm{misfit}} = 1.7$ for JFM mean that on average the misfit was larger than the uncertainty of the original data by 50 % to 70 %. However, the geographic distribution shows that large misfits were restricted to certain regions (e.g., subject to large variations due to upwelling or sea-ice cover) and maybe due to deviations between near-by sediment core locations.

We deliberately made use of the separate summer and winter temperature reconstructions based on the faunal and floral transfer function technique. This technique may not provide fully independent seasonally resolved NSST reconstructions (cf. Mix et al., 2001; Morey et al., 2005) but partly reflect the seasonal NSST structure of the calibration data set (Kucera et al., 2005b), as indicated by the very high correlation (r≈0.94) between the seasonal reconstructions and the winter and summer NSST in the calibration data sets (Kucera et al., 2005b). However, we are confident that some information on the amplitude of the seasonal cycle may still be inferred from microfossil abundances using the faunal and floral transfer function technique as long as both warmth- and cold-loving species are present and no-analog situations are avoided.

As detailed in Table 3, the recent studies by Annan and Hargreaves (2013), Kurahashi-Nakamura et al. (2017) and Tierney et al. (2020) use different data sets, models and methods. They all involve one or several dynamic models. For example, Kurahashi-Nakamura et al. (2017) use the method-of-Lagrange-multipliers or "adjoint method" (Wunsch, 1996) in combination with a particular ocean general circulation model (MITgcm). Given its physics and paramaterizations, the resulting field is dynamically consistent with the model, however, it also reflects its structural uncertainty, for example, as evident in the weak cooling or even warming near the eastern boundaries in Fig. 7, coastal upwelling systems cannot be resolved by a coarse-resolution ocean model. On the other hand, it shows a shift in the subtropical front at about 30° latitude in either hemisphere that is not seen in any of the other reconstructions. In contrast, our reconstruction is based on a statistical model, which makes fewer assumptions on how two data points are connected to each other, but it also lacks dynamically consistent constraints.

This may explain why our results indicate a slight warming in the tropical Pacific Ocean caused by the interpolation of a few and uncertain data points, in contrast to the dynamic models that induce a cooling comparable to that in the tropical Atlantic Ocean.

The reconstruction by Tierney et al. (2020) is based on a different data set that consists of geochemical proxies only and is combined with a particular coupled climate model (iCESM1.2) using an "off-line" data assimilation method. It yields a larger, more homogeneous cooling, except for the high southern latitudes, in which the Pacific sector cools more than the Atlantic sector.

Regarding the Mediterranean Sea, in the coarse reconstruction by Annan and Hargreaves (2013) it is represented as two separated "sub-seas", while it is completely missing in the reconstruction by Kurahashi-Nakamura et al. (2017). The "off-line" data assimilation method by Tierney et al. (2020) yields a homogeneous result. It seems that the GLOMAP reconstruction is the only one that can properly present the Mediterranean Sea, in terms of spatial resolution of the underlying finite-element mesh as well as in taking into account the available data (cf. Fig. 4 and Fig. 6).

When comparing the area-weighted regional anomalies in Table 4, we find that our results on the global and tropical ocean cooling are in the range of those that are also based on the MARGO faunal and floral assemblages, but lower than in the reconstruction by Tierney et al. (2020) based on geochemical proxies. One reason may be the use of a different reference temperature data set: Inherently, the reconstructions by Annan and Hargreaves (2013) and Kurahashi-Nakamura et al. (2017) as well as our reconstruction use the annual-mean temperature at a depth of 10 m from the World Ocean Atlas 1998 (WOA, 1998) as a reference. Tierney et al. (2020), however, use a Late Holocene (4,000–0 years before present) reconstruction as a reference, which may produce different anomaly patterns. Another reason may be the use of different proxies.

At the level of individual ocean sediment cores, the best resolved alkenone-based NSST estimates from the central Pacific Ocean show an NSST change between 1.2 °C and 2 °C (Broccoli and Marciniak, 1996; Prahl et al., 1989; Lee et al., 2001; de Garidel-Thoron et al., 2007). From a number of studies using Mg/Ca as well as alkenones, Lea (2004) find a tropical cooling at the LGM by (2.8±0.7) °C. Leduc et al. (2017) summarize the results of the Sensitivity of the Tropics (SENSETROP) working group, which after the incorporation of high-quality records and a thorough quality control obtains a cooling of the low latitudes during the LGM by (2.3±0.8) °C and (2.4±0.8) °C for alkenone- and Mg/Ca-based NSST estimates, respectively. Tierney et al. (2020) obtain a very similar mean tropical cooling by 2.5 °C (2.2 °C to 2.8 °C, 95 % confidence interval) from the NSST proxies on their own. These values are larger than the estimates by CLIMAP, MARGO and Annan and Hargreaves (2013) by up to a degree, but not as large as the early estimate from corals by Guilderson et al. (1994) of about 5 °C (see also the summary in Manabe and Broccoli, 2020).

A possible reason for the difference between faunal and floral proxies on the one hand and geochemical proxies on the other hand is the so-called no-analog problem: There may be assemblages in the fossil record that do not have a counterpart in the modern calibration data set. However, the MARGO project in particular carefully dealt with this problem in a number of ways. For example, Gersonde et al. (2005) discard all samples with no analogs (dissimilarity > 0.25) and when the majority of the samples in the LGM interval has no analogs, the estimated quality level is downgraded to 3. Kucera et al. (2005b) combine three methods (Artificial Neural Networks – ANN, Revised Analog Method – RAM, Maximum Similarity Technique

– SIMMAX) in a multi-technique approach that facilitates a test of the robustness of NSST estimates and provides a means to identify potential no-analog conditions or faunas.

Another possible reason are low sedimentation rates, in particular in the tropical Pacific Ocean. However, in comparing different proxies, de Garidel-Thoron et al. (2007) find in a well-resolved alkenone record from the western Pacific Warm Pool a cooling by 1 °C to 2 °C only, too, which would be consistent with the MARGO results.

    While faunal and floral assemblages offer some advantages over geochemical proxies regarding spatial coverage, their potential to provide a seasonal reconstruction and internal consistency, they are not without issues. For example, there are large
discrepancies between the NSST reconstructions from foraminiferal and dinocyst assemblages in the Nordic Seas (de Vernal et al., 2006). The apparently warm signal recorded by dinocyst assemblages may be due to long-distance lateral transport (de Vernal et al., 2006). Further issues with dinoflagellates are their overwintering in a cyst phase and broad tolerances for temperature (Dale, 2001). Coccolithophores and alkenones are also prone to long-distance lateral transport (Rühlemann and Butzin, 2006), whereas foraminifera-based proxies have the advantage that their signal carriers drop relatively quickly to the
sediment (Takahashi and Be, 1984). In addition, using alkenones for NSST reconstructions in high latitudes may be problematic because of the low sensitivity of the calibration of alkenones at low temperature (Conte et al., 2006), the possibility of redeposition of old and warm signal carrying alkenones with particulate matter originating from the glaciated continental margins and once more the influence of alkenones transported by currents from warmer areas into the polar regions (e.g., Bendle and Rosell-Melé, 2004; Filippova et al., 2016).

Since foraminiferal assemblages are usually dominated by species adapted to the environment of the overlying water column (Morey et al., 2005), we consider the temperature estimation to be more robust against the expatriation of single shells that can affect proxies measured on monospecific samples. Finally, proxies based on the chemistry of shells of living organisms suffer from the inherent problem that the environmental sensitivity of that organism biases the recording of the proxy (Mix, 1987; Fraile et al., 2009). The transfer function method does not have this problem since it actually uses the environmental sensitivity
of the fauna.

    From this discussion we conclude that assemblages of living foraminifera faithfully record environmental conditions. However, there are a number of challenges in interpreting the fossil record, in attributing the result to a certain season and water depth and estimating a global cooling from the still sparse and irregularly spaced data set:

1. *Change in seasonality*: Ravelo et al. (1990) demonstrate that in the equatorial Atlantic Ocean faunal assemblages do not
respond primarily to NSST, but rather to thermocline and seasonality changes. In fact, using a foraminifera model, Fraile et al. (2009b) show that during the LGM, the maximum production of subtropical as well as high-latitude foraminifera is generally shifted towards a warmer season of the year. For tropical species the change in seasonality did not produce an important temperature bias, because the amplitude of the annual cycle is relatively low.

2. *Change in thermal structure*: Telford et al. (2013) indeed provide evidence that planktonic foraminifera assemblages
can be more sensitive to subsurface temperatures than the 10-m NSST that they are usually calibrated against, e.g., as in MARGO. They conclude that reconstructions of the 10-m NSST are likely to be biased, with the sign and magni-

tude of the bias varying regionally, but probably causing a warm bias in the tropical North Atlantic Ocean. However, foraminifera-based reconstructions for other ocean basins still need to be assessed.

3. *Sampling bias*: The majority of NSST estimates comes from the continental margins and exhibits systematic deviations
from the open ocean that are related to gyre circulation [cf. Judd et al. (2020) and references therein]. For example, eastern continental margins are dominated by coastal upwelling of cold sub-surface water and radiative cooling and less sensitive to surface cooling and hence are prone to yield a reduced glacial-interglacial contrast. Judd et al. (2020) also point out that data assimilation methods may be helpful in overcoming this spatial bias, provided that the models capture the zonal heterogeneity in temperature due to coastal dynamics.

Taken at face value, and using the same linear relationship by Schneider von Deimling et al. (2006, their Fig. 6) as the MARGO Project Members (2009), our result of a mean change for the tropical Atlantic Ocean of (-2.1±0.7) °C [here for consistency with Schneider von Deimling et al. (2006) taken between 20° S and 20° N] would correspond to an equilibrium climate sensitivity of (1.5±1.0) °C. This is at the low end of the classical range from 1.5 °C to 4.5 °C considered to be likely by Collins et al. (2013, Box 12.2), and it is even lower than the estimate by the MARGO Project Members (2009) of (2.3±1.3) °C.
However, these values need to be put in a proper perspective.

On the one hand, our discussion shows that because of changes in seasonality and thermal structure as well as uneven sampling, our estimate of the global and tropical NSST decrease based on the MARGO faunal and floral assemblages is likely to biased towards lower values. While at present this is difficult to quantify, the geochemial proxies at the tropical sediment core sites investigated by Lea (2004), Leduc et al. (2017) and Tierney et al. (2020) indeed suggest a cooling of around 2.5 °C,
larger by about 0.5 °C to 1 °C. Whether it should be even larger as proposed by Tierney et al. (2020) based on their off-line data assimilation still needs to be independently confirmed. In any case, according to the simple linear relationship by Schneider von Deimling et al. (2006), a larger cooling would also imply a higher equilibrium climate sensitivity.

On the other hand, in view of the recent comprehensive review by Sherwood et al. (2020), estimating the equilibrium climate sensitivity is a very challenging task and simply using a linear relationship derived from a single model does not seem to be an
405 adequate approach. Combining three lines of evidence (observed climate processes, historical climate changes and paleoclimate changes), these authors conclude that the equilibrium climate sensitivity is likely in the slightly narrower range between 2.6 °C and 4.1 °C (at a 66 % level of confidence). One of the two periods that they consider for paleoclimate evidence is the LGM, for which they assume a global surface air temperature decrease of about 3 °C to 7 °C with respect to the pre-industrial period (Sherwood et al., 2020, Section 5.2.1), according to the authors a value inferred from observations at low latitudes (Sherwood
et al., 2020, Section 5.2.4).

Based on the reconstruction by Annan and Hargreaves (2013), who apply multiple linear regression to the PMIP2 ensemble of climate models (Braconnot et al., 2007), a decrease in sea-surface temperature in the tropical Atlantic Ocean of 1.5 °C corresponds to a decrease of global surface air temperature by (4.0±0.8) °C (95 % confidence interval). Hence we expect the tropical cooling between 1.0 °C and 1.5 °C in our and the original MARGO reconstruction to be consistent with a decrease
of global surface air temperature by about 3 °C, which in the analysis by Sherwood et al. (2020) would contribute to their

lower limit on the equilibrium climate sensitivity of 2.6 °C, while a tropical cooling of around 2.5 °C or larger as inferred from reconstructions based on geochemical proxies would place the equilibrium climate sensitivity more in the middle of their range.

## 5 Conclusions

In summary:

- – We demonstrated that the Data-Interpolating Variational Analysis (DIVA Troupin et al., 2012) method can be applied to irregularly spaced data that are much sparser than current oceanographic observations, at a computational cost that is orders of magnitude lower than for the assimilation of the data in a coupled climate or ocean model.

- – Consequently, using DIVA, we derived an internally consistent climatology of the monthly NSST and sea-ice extent
during the LGM on a global regular $1° \times 1°$ grid from the MARGO microfossil assemblages (Kucera et al., 2005a) and a number of sea-ice reconstructions.

- – Based on this gridded climatology, we confirmed that the longitudinal and meridional NSST differences were likely to be greater than today.

- – Using the uncertainty estimate provided by DIVA as a weight, we calculated global and regional averages, quantified the
meridional NSST differences, estimated the respective uncertainties, and, for example, obtained a cooling of the global ocean by (1.7±0.1) °C and of the tropical ocean by (1.2±0.3) °C.

- – From a review of processes that affect faunal and floral assemblages we concluded that they are faithful recorders of the actual environmental conditions, but that there are a number of challenges in interpreting their fossil record, especially in attributing the local sedimentary imprint to a particular season and water depth.

- – Hence anticipated changes in seasonality and thermal structure and a spatial sampling bias, as well as a comparison to geochemical proxies at comparable sites, let us conjecture that results on the global and tropical cooling based on faunal and floral assemblages are likely to be biased towards lower values by at least 0.5 °C to 1 °C.

- – This implies that estimates of equilibrium climate sensitivity derived from estimates global and tropical cooling based on faunal and floral assemblages and taken at face value tend to be on the low side, too, while estimates based on
geochemical proxies would place it more in the middle of the range given in the recent review by Sherwood et al. (2020).

## 6 Outlook

We expect the Glacial Ocean Map (GLOMAP), in terms of the gridded field and error estimate provided by DIVA, to prove useful in several ways, for example, in evaluating coupled climate models and forcing AGCMs in simulations of the climate

of the LGM, or in first smoothing and spreading the original sparse data before using it in constraining an inverse model (cf.
Marchal and Curry, 2008). Regarding the first application, we plan to use water isotopes as a tool to compare the performance of different AGCMs, using our gridded GLOMAP NSST climatology as a common boundary condition. This way we can on the one hand avoid the propagation of the simulated SST bias in coupled climate models, and on the other hand we can isolate the impact of the ocean feedback on the simulated distributions of water isotopes over land, ice and ocean (e.g., Werner et al., 2018). We also plan to extend our method to $\delta^{18}$O from fossil calcite shells of planktonic foraminifera. A combined
reconstruction of NSST, sea ice coverage and the inferred $\delta^{18}$O of seawater may be used for an enhanced evaluation of coupled climate models. Regarding future additions to the MARGO database, we hope for an improved coverage of the interior oceans, particularly the tropical Pacific Ocean and the Northwestern Atlantic Ocean. Our application of the DIVA method may be further refined by, for example, including advection by surface currents. Improving the attribution of fossil faunal and floral assemblages to a certain season or water depths would, however, require a more complex approach, for example, by combining
a coupled climate model with a planktonic foraminifera model such as PLAFOM in Kretschmer et al. (2018).

*Data availability.* The GLOMAP gridded climatology of monthly LGM NSST anomalies (including their uncertainties) and monthly estimates of LGM sea-ice extent are available through PANGAEA (https://doi.pangaea.de/10.1594/PANGAEA.923262). It may be updated when new reconstructions become available.

*Author contributions.* AP and MW formulated the research goals and aims. AP designed the methodology, implemented the computer code,
analyzed the data and visualized the results. SM and RS verified the use and interpretation of the paleo data. AP prepared the manuscript with contributions from all co-authors.

*Competing interests.* The authors declare that they have no conflict of interest.

*Acknowledgements.* We are grateful to Jessica Tierney and an anonymous referee for their constructive reviews that helped us to greatly improve our manuscript. This work was supported by the German Federal Ministry of Education and Research (BMBF) as a Research for
Sustainability initiative (FONA) through the PalMod project (FKZ: 01LP1511D) as well as by the DFG Research Center/Center of Excellence MARUM – "The Ocean in the Earth System".

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

| Ocean | DIVA | Kriging | Levitus |
|-------|------|---------|---------|
| Atlantic | 1.33 | 1.29 | 1.40 |
| Pacific | 1.16 (1.11) | 1.19 (1.15) | 1.75 |
| Indian | 0.73 | 0.93 | 1.04 |
| Arctic | 1.30 | 3.52 (1.69) | 1.84 |
| | | | |
| Global | 1.16 (1.14) | 1.22 (1.15) | 1.56 |

**Table 1.** Root-mean square annual-mean NSST differences between the interpolated and observed fields at all WOA (1998) unanalyzed data locations, binned into $2°$ longitude/latitude squares. Differences in parentheses arise if all NSST values below -1.8$°$ C (taken as the freezing point) are set to this value in the analyzed field.

**Table 2.** Global and regional averages and meridional differences of NSST anomalies (LGM - modern) based on data gridded with DIVA and weighted by their uncertainties.

| GLOMAP regional averages, meridional differences and uncertainties | | |
|---|---|---|
| Region | Average | Uncertainty |
| Global ocean | -1.7 | 0.1 |
| Global tropical ocean (30° S - 30° N) | -1.2 | 0.3 |
| Northern tropical ocean (0° - 30° N) | -0.9 | 0.4 |
| Southern tropical ocean (30° S - 0°) | -1.4 | 0.4 |
| Tropical meridional difference (north - south) | 0.4 | 0.6 |
| Tropical Atlantic Ocean (30° S - 30° N) | -1.8 | 0.6 |
| Tropical Atlantic Ocean (20° S - 20° N) | -2.1 | 0.7 |
| Northern Tropical Atlantic Ocean (0° - 30° N) | -1.6 | 0.8 |
| Southern Tropical Atlantic Ocean (30° S - 0°) | -2.0 | 0.8 |
| Tropical Atlantic meridional difference | 0.4 | 1.2 |
| Northern North Atlantic Ocean (> 45° N) | -3.1 | 0.2 |
| Southern South Atlantic Ocean (< -30° S) | -1.4 | 0.3 |
| Atlantic meridional difference (north - south) | -1.7 | 0.3 |

All temperature anomalies and uncertainties in units of °C

| Study | LGM Data | Model(s) | Method |
|---|---|---|---|
| Annan and Hargreaves (2013) | global annual-mean NSST (MARGO, 2009) and SAT (Bartlein et al., 2011; Shakun et al., 2012) | PMIP2 ($\approx 1°$ to $\approx 3°$) | multiple linear regression |
| Kurahashi-Nakamura et al. (2017) | global annual-mean NSST (MARGO, 2009), Atlantic benthic $\delta^{18}O$ (Marchal and Curry, 2008) and $\delta^{13}C$ (Hesse et al., 2011) | MITgcm ($\approx 3°$) | method-of-Lagrange-multipliers/adjoi |
| Tierney et al. (2020) | global annual-mean NSST, various geochemical reconstructions ($U_{37}^{K'}$, $TEX_{86}$, Mg/Ca, $\delta^{18}O$) | iCESM1.2 ($\approx 1°$) | off-line data assimilation |
| GLOMAP (this study) | global seasonal NSST (MARGO, 2009, faunal and floral assemblages), various sea-ice reconstructions | statistical ($\approx 1°$) | variational inverse method (DIVA) |

**Table 3.** Comparison of global gridded climatologies of the ocean surface during the LGM. With respect to the models employed, the approximate horizontal grid resolution is given in brackets. The NSST results from the multiple linear regression by Annan and Hargreaves (2013) are provided on a regular grid with a constant resolution of $5°$. For more details, see the respective study.

| Region | CLIMAP | GLAMAP | AH2013 | K2017 | T2020 | GLOMAP |
|---|---|---|---|---|---|---|
| Global Ocean | -1.5 | -1.8 | -2.1 | -2.0 | -3.6 | -1.7 |
| Global tropical ocean (20° S - 20° N) | -0.9 | -1.2 | -1.5 | -2.1 | -3.4 | -1.0 |
| Northern tropical ocean (0° - 20° N) | -1.1 | -1.3 | -1.6 | -2.3 | -3.4 | -0.8 |
| Southern tropical ocean (20° S - 0°) | -0.7 | -1.1 | -1.4 | -1.8 | -3.5 | -1.2 |
| Tropical Atlantic Ocean (20° S - 20° N) | -1.6 | -2.8 | -2.1 | -2.4 | -3.7 | -2.1 |
| Northern Tropical Atlantic Ocean | -1.6 | -2.5 | -1.9 | -3.0 | -3.7 | -1.6 |
| Southern Tropical Atlantic Ocean | -1.5 | -3.2 | -2.2 | -1.7 | -3.7 | -2.6 |
| Northern North Atlantic Ocean (> 45° N) | -7.1 | -5.8 | -3.1 | -3.3 | -4.8 | -5.4 |
| Southern South Atlantic Ocean (< -30° S) | -1.9 | -2.4 | -2.4 | -2.6 | -2.2 | -2.5 |

**Table 4.** Comparison of global gridded climatologies of the ocean surface during the LGM in terms of the area-weighted regional anomalies of the annual-mean NSST (CLIMAP = CLIMAP Project Members (1981), GLAMAP = Sarnthein et al. (2003a), AH2013 = Annan and Hargreaves (2013), K2017 = Kurahashi-Nakamura et al. (2017), T2020 = Tierney et al. (2020) and GLOMAP = this study).

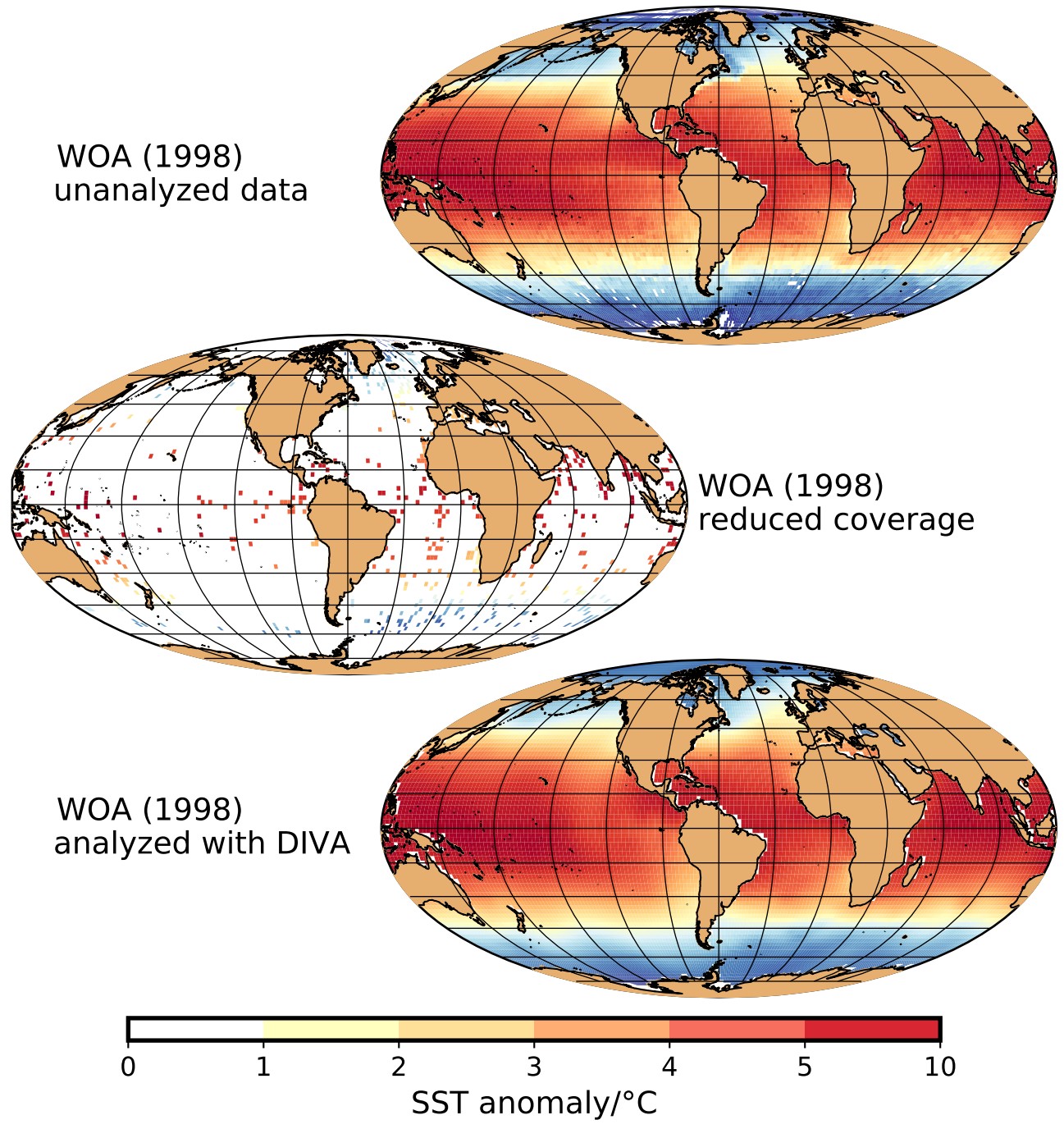

**Figure 1.** *Top:* Unanalyzed annual-mean temperature at a depth of 10 m from the World Ocean Atlas 1998 (WOA, 1998), binned into a regular grid with a constant resolution of 2° using the GMT program xyz2grd (Wessel and Smith, 1998). *Middle:* The same data, but after greatly reducing the coverage by keeping only those 2° squares that contain an ocean sediment core site from MARGO Project Members (2009). This is the input data set for testing the DIVA method (Troupin et al., 2012). *Bottom:* The result of using the DIVA method to interpolate the sparse test data to a complete regular grid with a constant resolution of 2°.

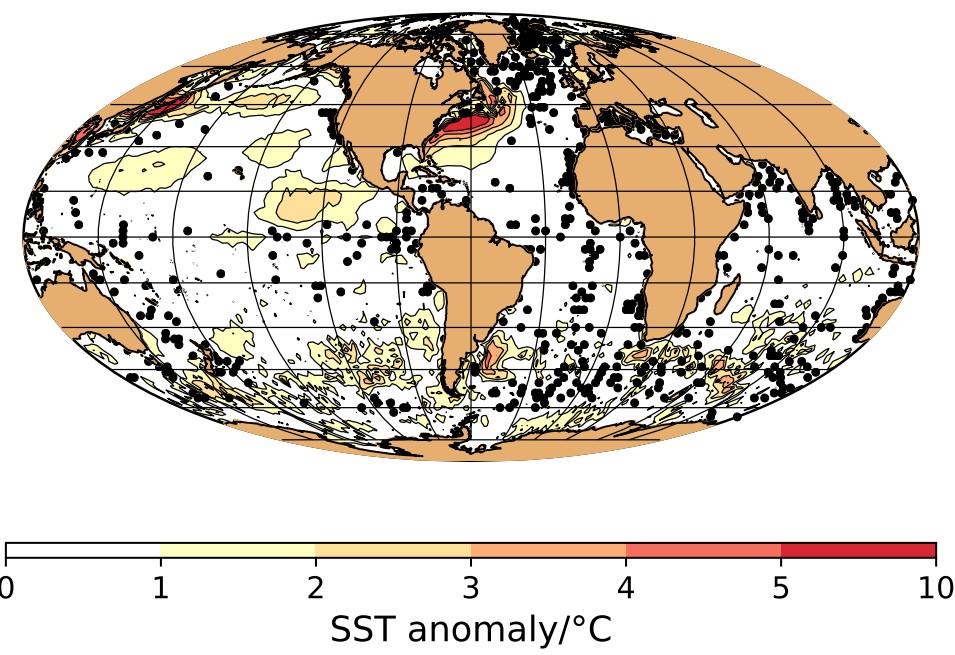

**Figure 2.** Absolute difference between the analyzed (using the DIVA method, Troupin et al., 2012) and unanalyzed (from the World Ocean Atlas 1998, WOA, 1998) annual-mean temperature at a depth of 10 m, shown as contour lines (for the contour intervals, see the color bar). In addition, the MARGO ocean sediment core sites are depicted as black circles (MARGO Project Members, 2009).

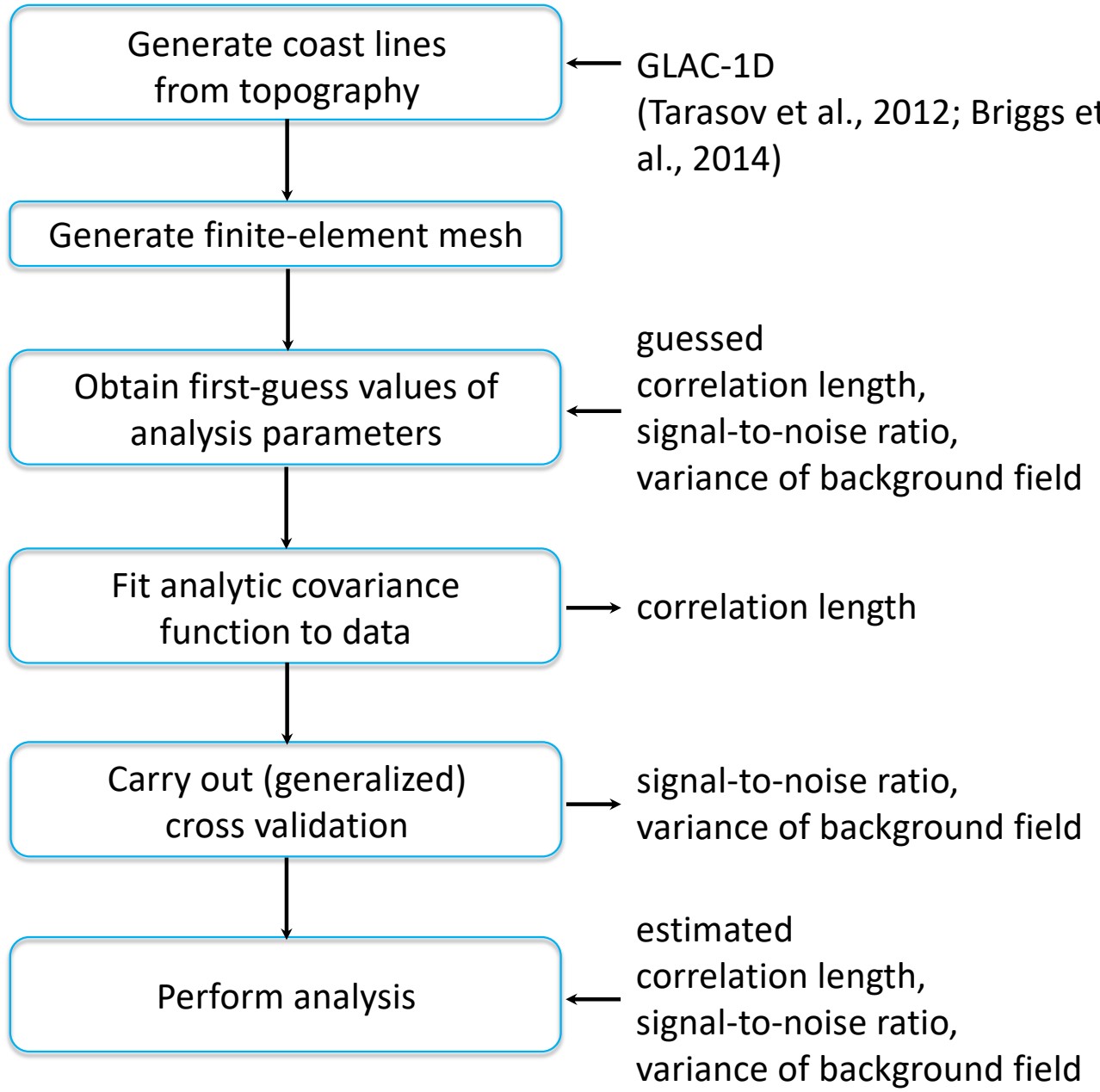

**Figure 3.** General workflow of the DIVA (Data-Interpolating Variational Analysis) method (Troupin et al., 2012).

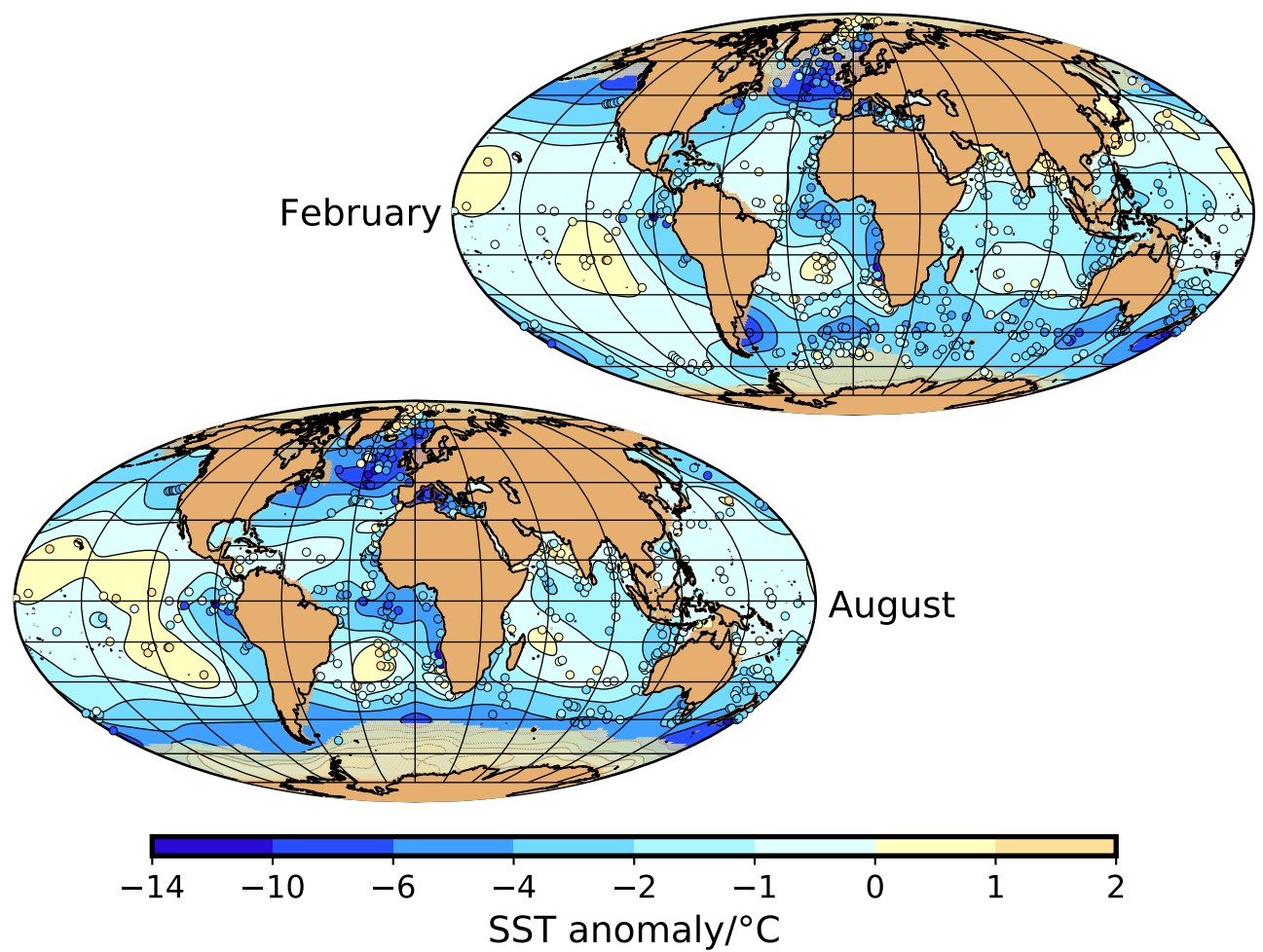

**Figure 4.** Analyzed LGM NSST anomalies for February and August (contour map) and data points (colored circles). In total, there are 600 data points for February and 464 data points for August (without the data points that are assumed to be covered by sea ice). The anomalies are relative to WOA (1998). The yellow-brownish areas close to Antarctica and in the Arctic indicate the LGM sea-ice masks based on the selected LGM sea-ice reconstructions.

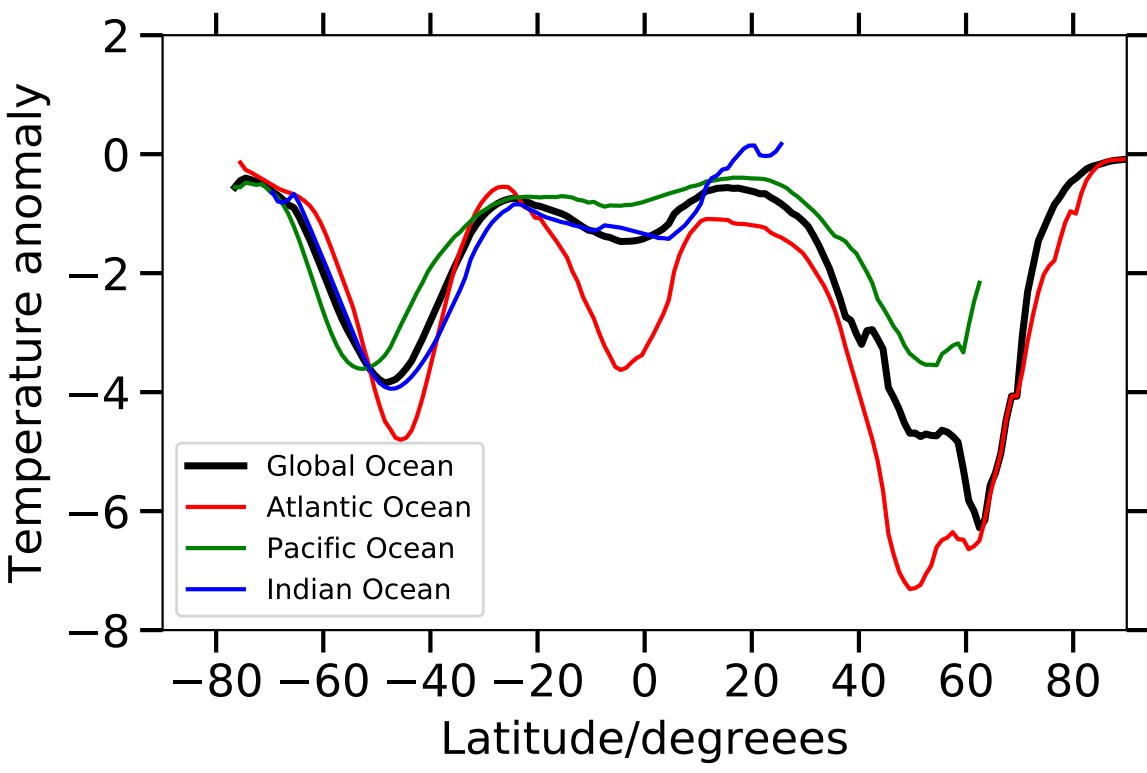

**Figure 5.** Zonally-averaged annual-mean NSST anomalies for the global ocean, Atlantic Ocean, Pacific Ocean and Indian Ocean.

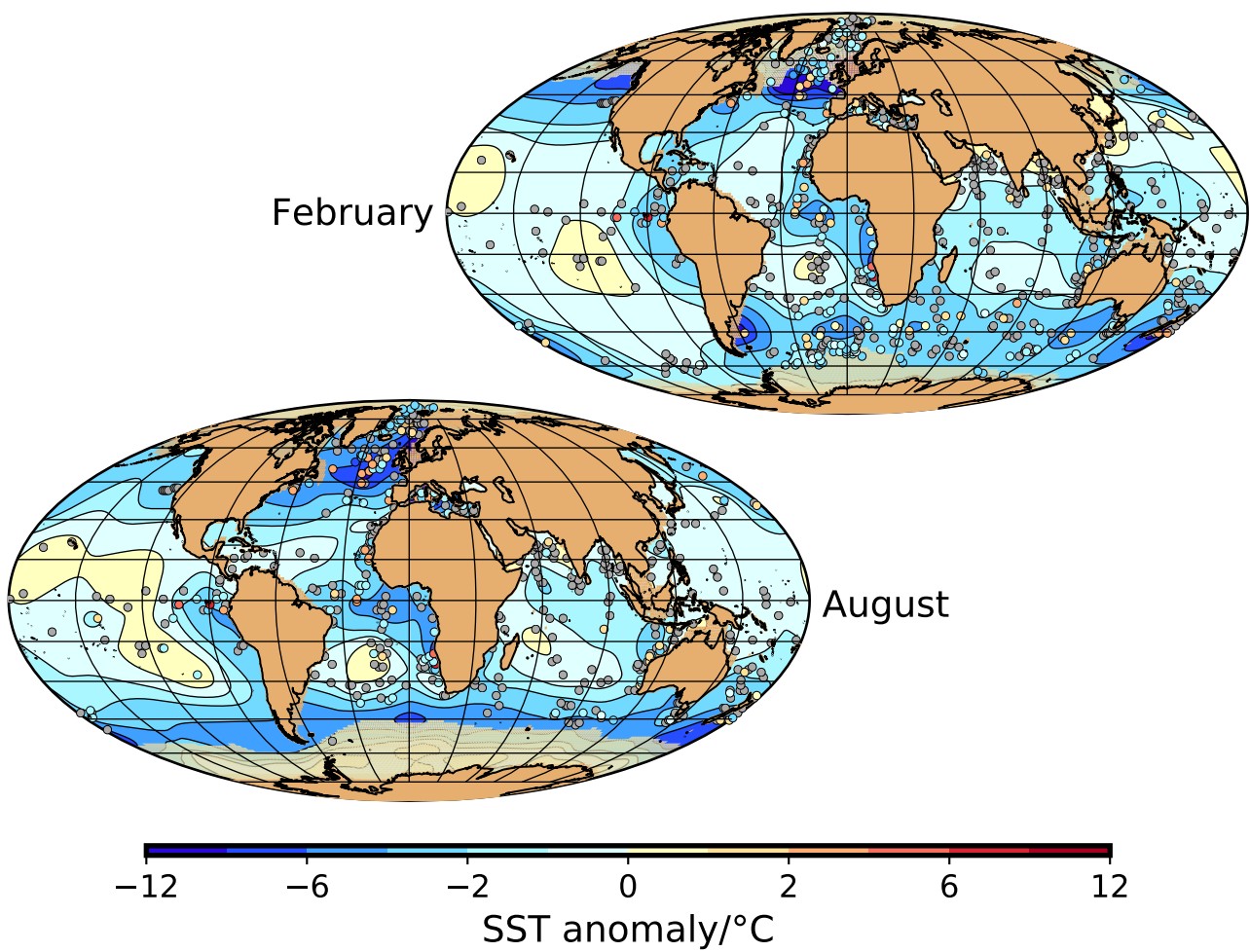

**Figure 6.** Analyzed NSST anomalies for February and August (contour map) and differences between the gridded values and the block-averaged original data values (colored circles – differences smaller than the uncertainty of the data are shown in dark grey).

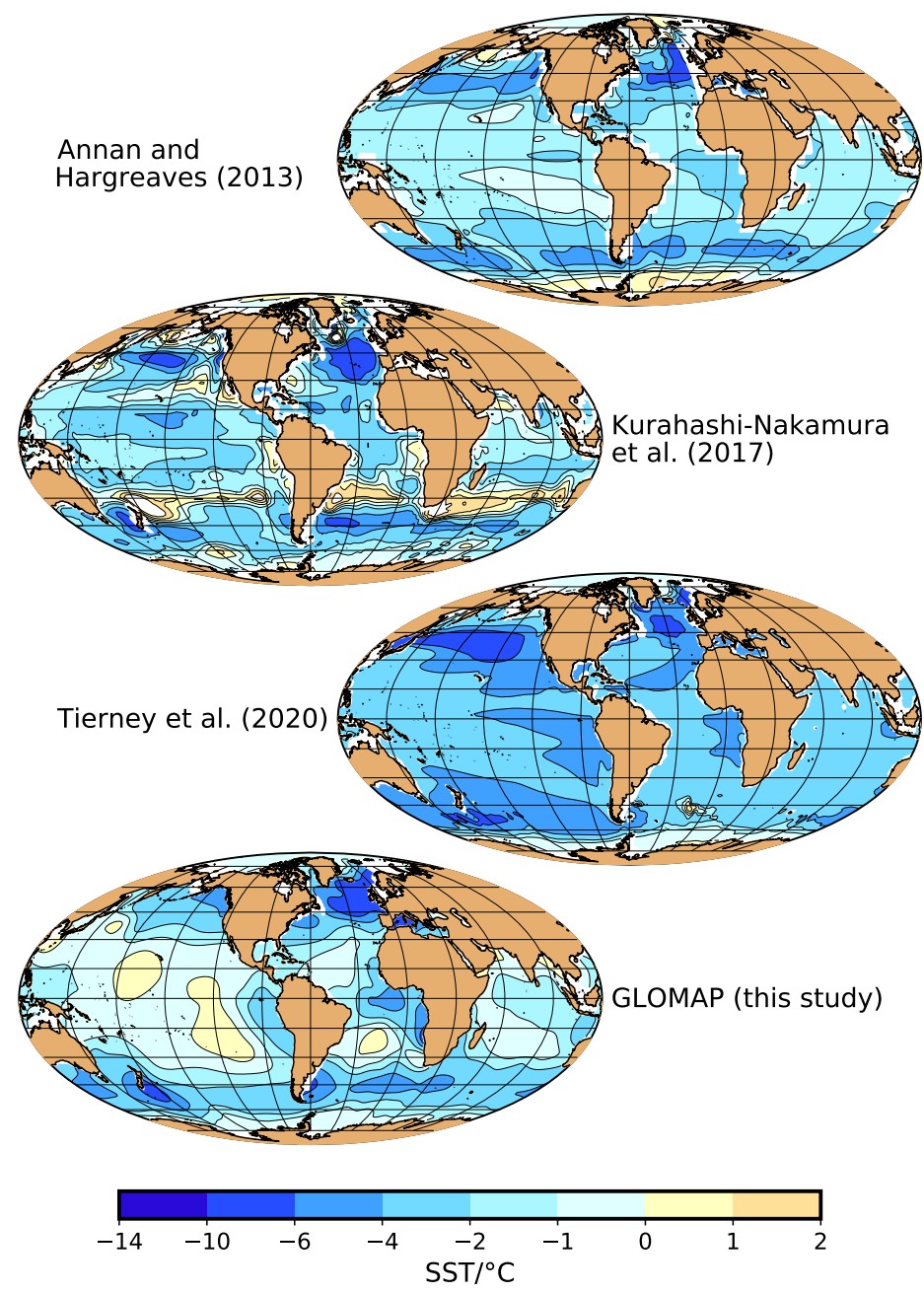

**Figure 7.** Annual-mean sea-surface temperature anomalies for the LGM according to the reconstructions by Annan and Hargreaves (2013), Kurahashi-Nakamura et al. (2017) and Tierney et al. (2020) on the one hand and GLOMAP on the other hand.

**Appendix A:  Maps of coastlines and finite-element meshes, monthly NSST anomalies and their estimated**

**uncertainties**

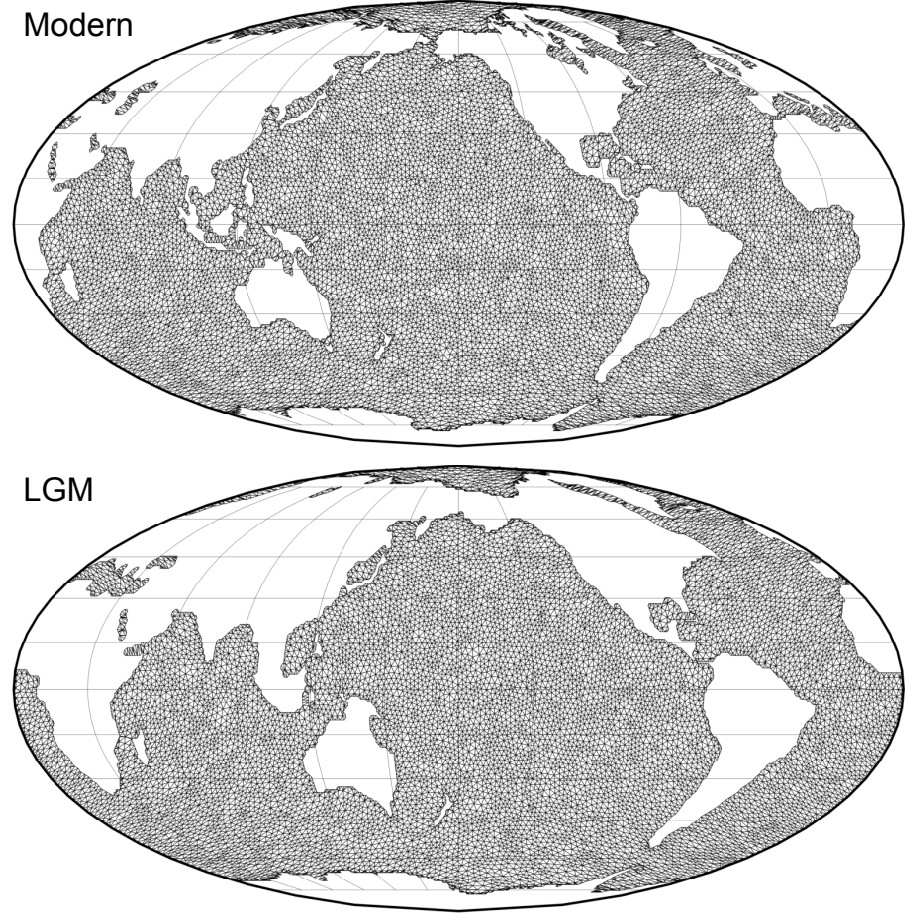

**Figure A1.** Coastline contours and finite-element meshes for the "original grids". *Top:* for the WOA test of the DIVA method, centered on 210° W (based on the modern bottom depth assigned to each 1° square by Garcia et al., 2019). *Bottom:* for the GLOMAP analysis, centered on 180° W (based on the LGM topography GLAC-1D by Tarasov et al., 2012; Briggs et al., 2014).

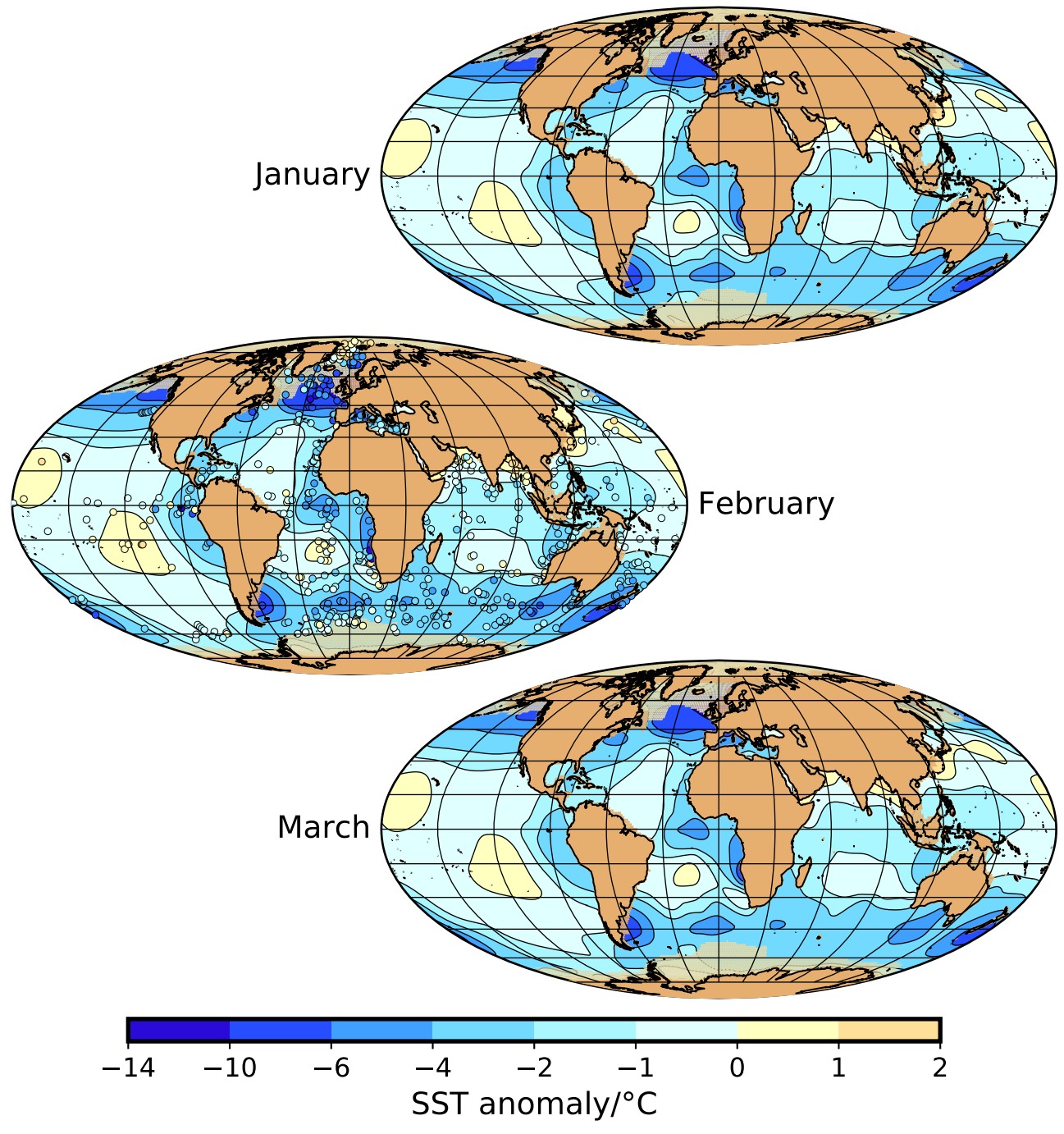

**Figure A2.** Sea-surface temperature anomaly (contour map) and sea-ice extent (yellow-brownish areas close to Antarctica and in the Arctic) for January, February and March. For February, we also show the MARGO reconstruction at the sediment core locations (MARGO Project Members, 2009).

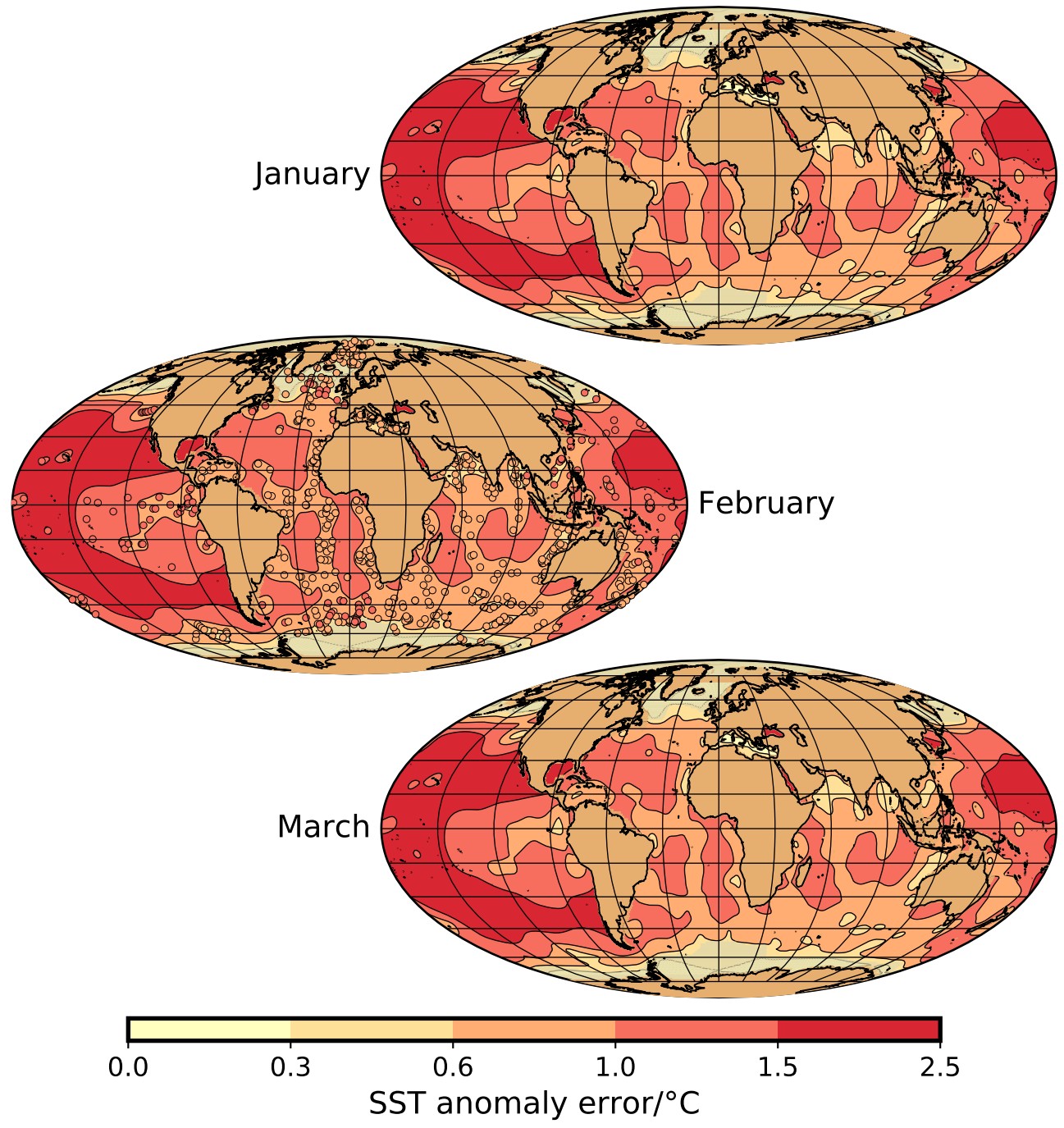

**Figure A3.** Uncertainty of NSST anomaly (contour map) and sea-ice extent (yellow-brownish areas close to Antarctica and in the Arctic, cf. Fig. A2) for January, February and March. For February, we also show the error of the reconstruction at the sediment core locations as estimated by the MARGO Project Members (2009).

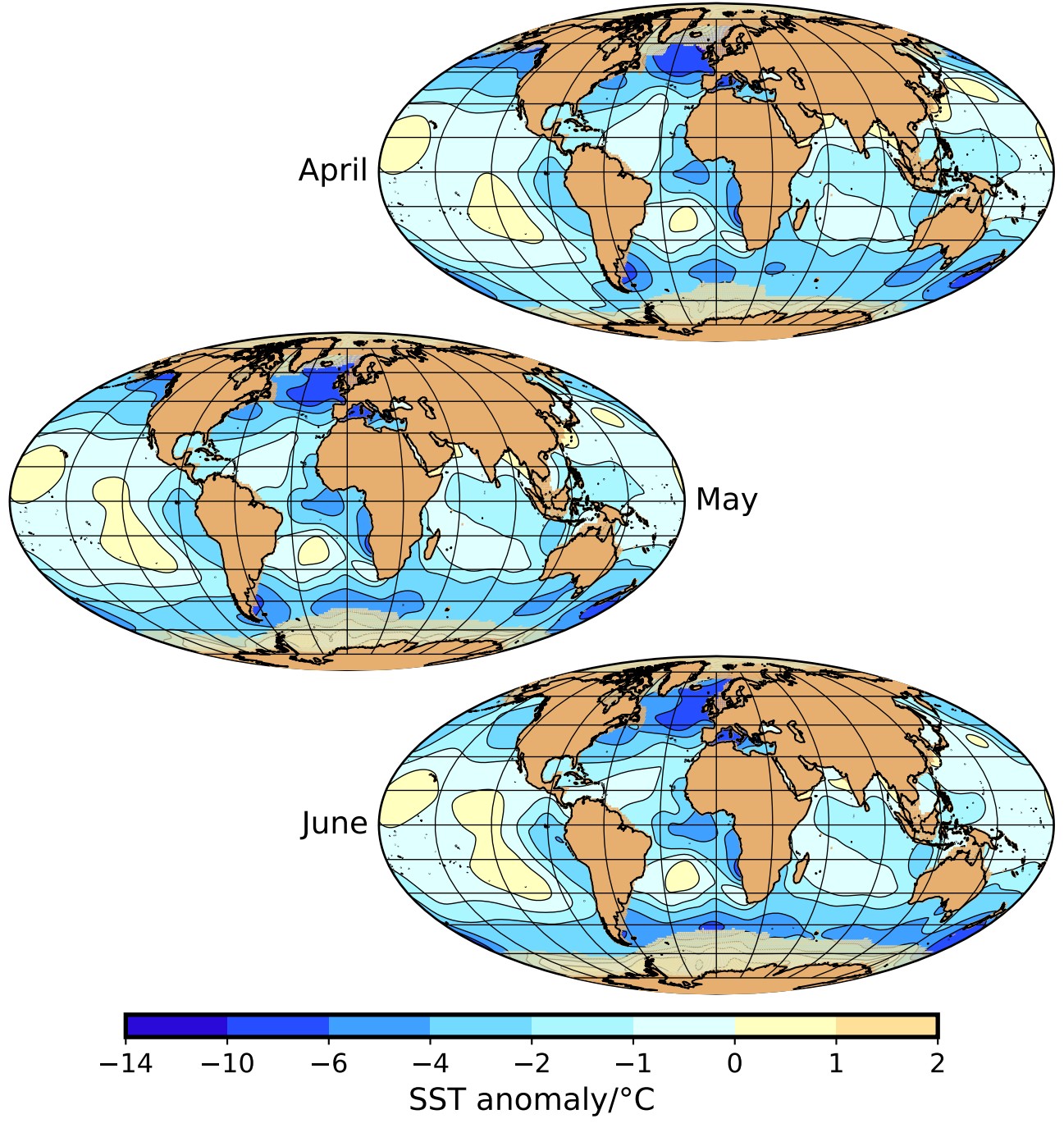

**Figure A4.** Sea-surface temperature anomaly (contour map) and sea-ice extent (yellow-brownish areas close to Antarctica and in the Arctic) for April, May and June

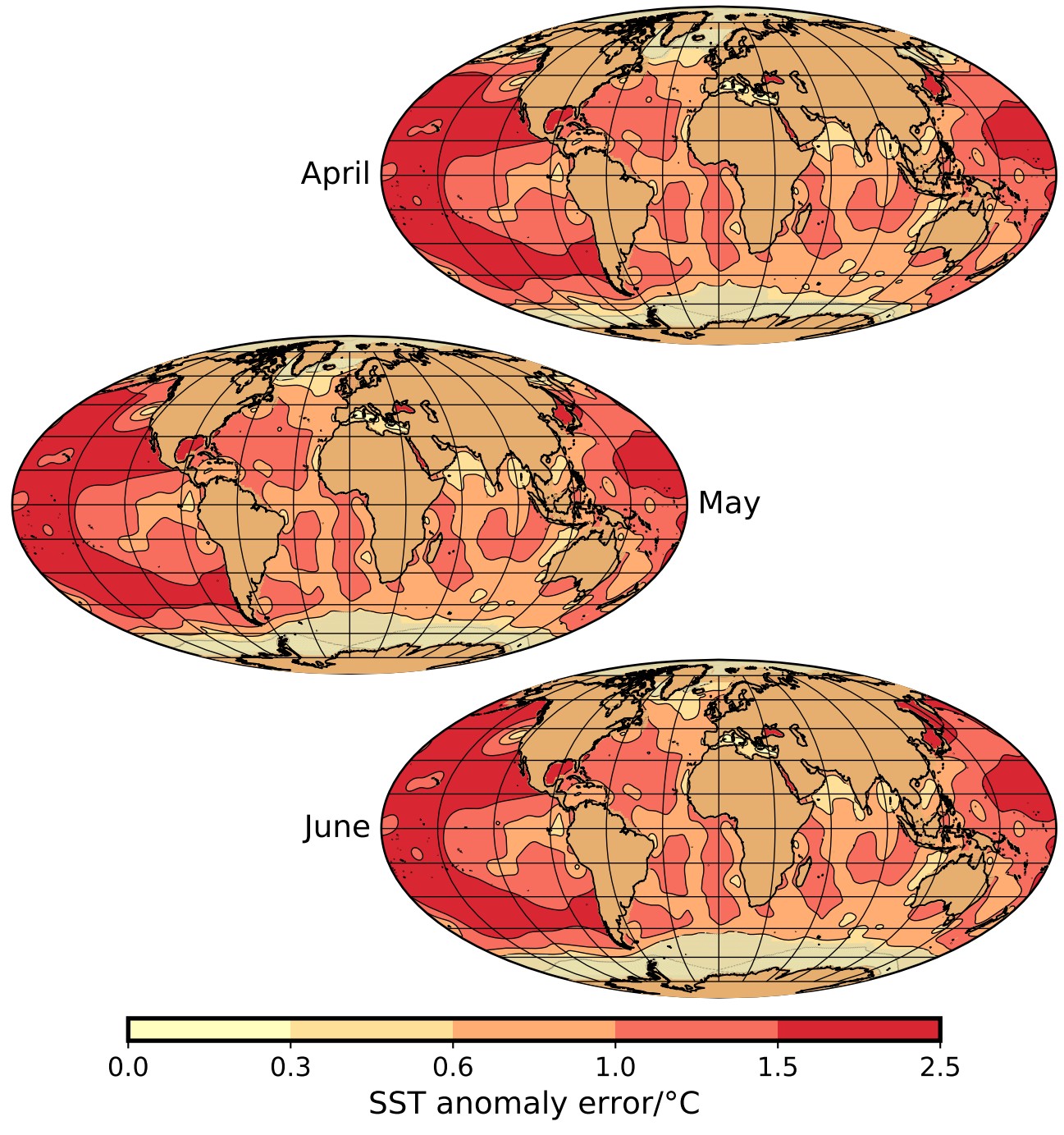

**Figure A5.** Uncertainty of NSST anomaly (contour map) and sea-ice extent (yellow-brownish areas close to Antarctica and in the Arctic, cf. Fig. A4) for April, May and June

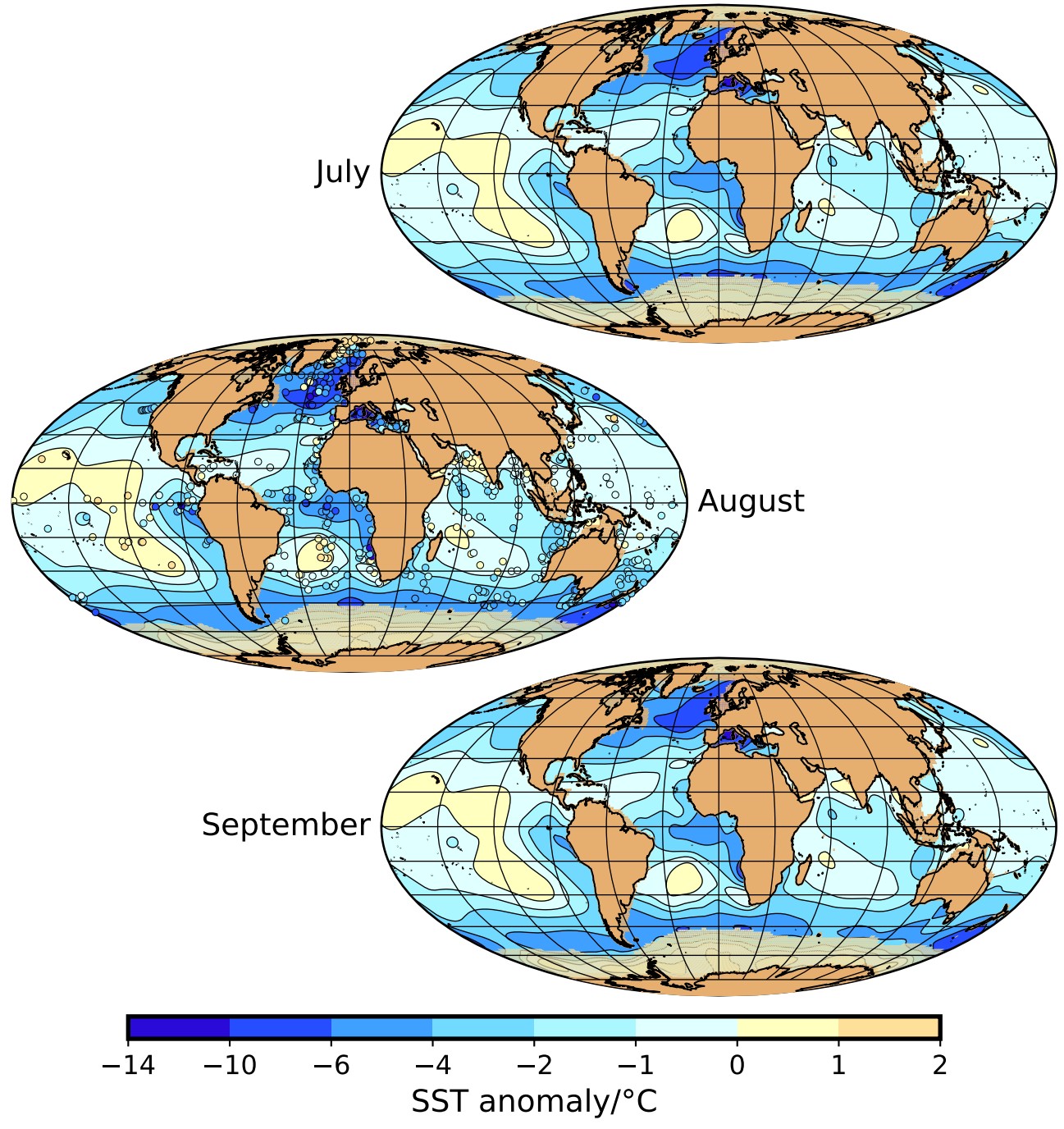

**Figure A6.** Sea-surface temperature anomaly (contour map) and sea-ice extent (yellow-brownish areas close to Antarctica and in the Arctic) for July, August and September. For August, we also show the MARGO reconstruction at the sediment core locations (MARGO Project Members, 2009).

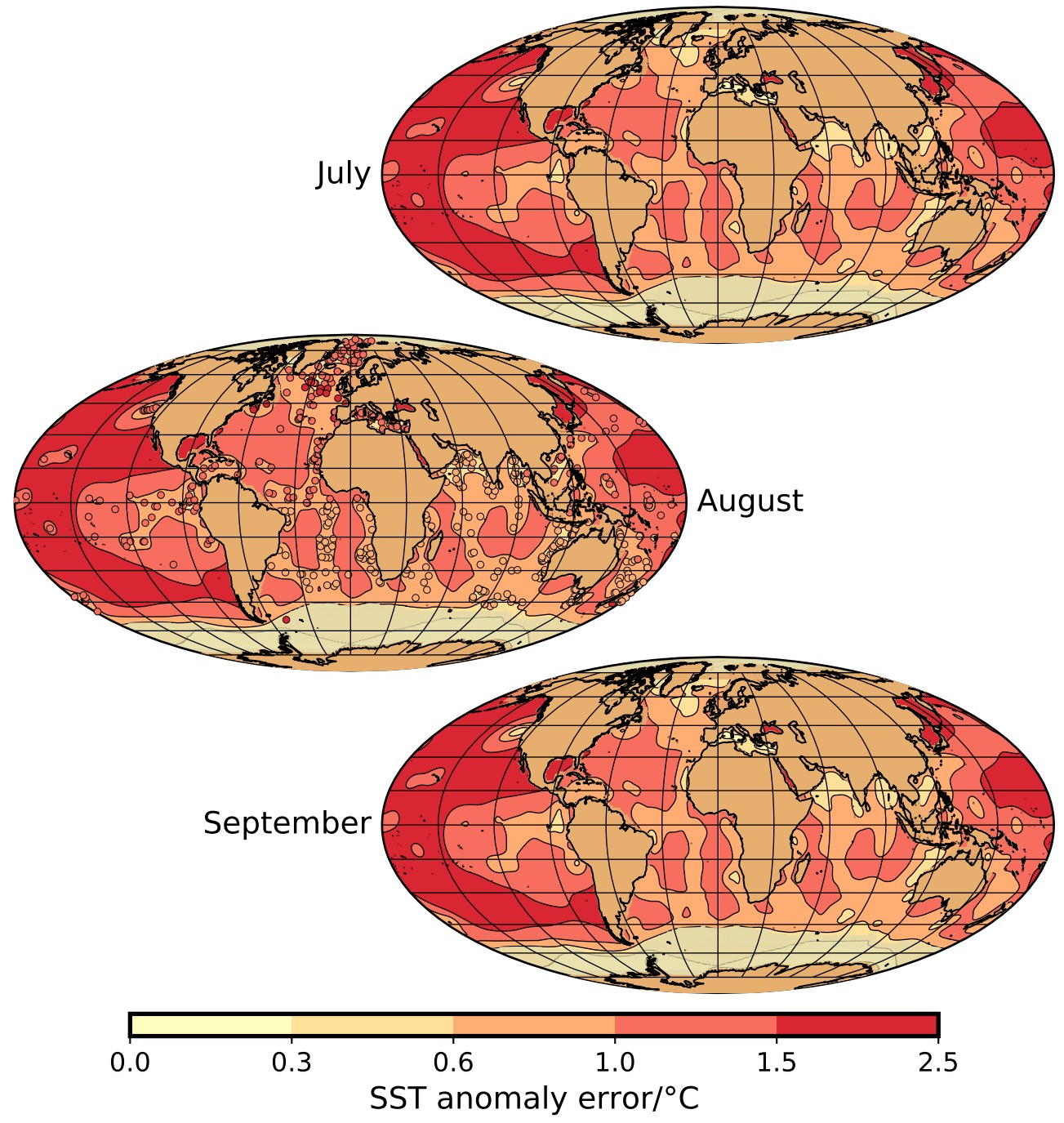

**Figure A7.** Uncertainty of NSST anomaly (contour map) and sea-ice extent (yellow-brownish areas close to Antarctica and in the Arctic, cf. Fig. A6) for July, August and September. For August, we also show the error of the reconstruction at the sediment core locations as estimated by the MARGO Project Members (2009).

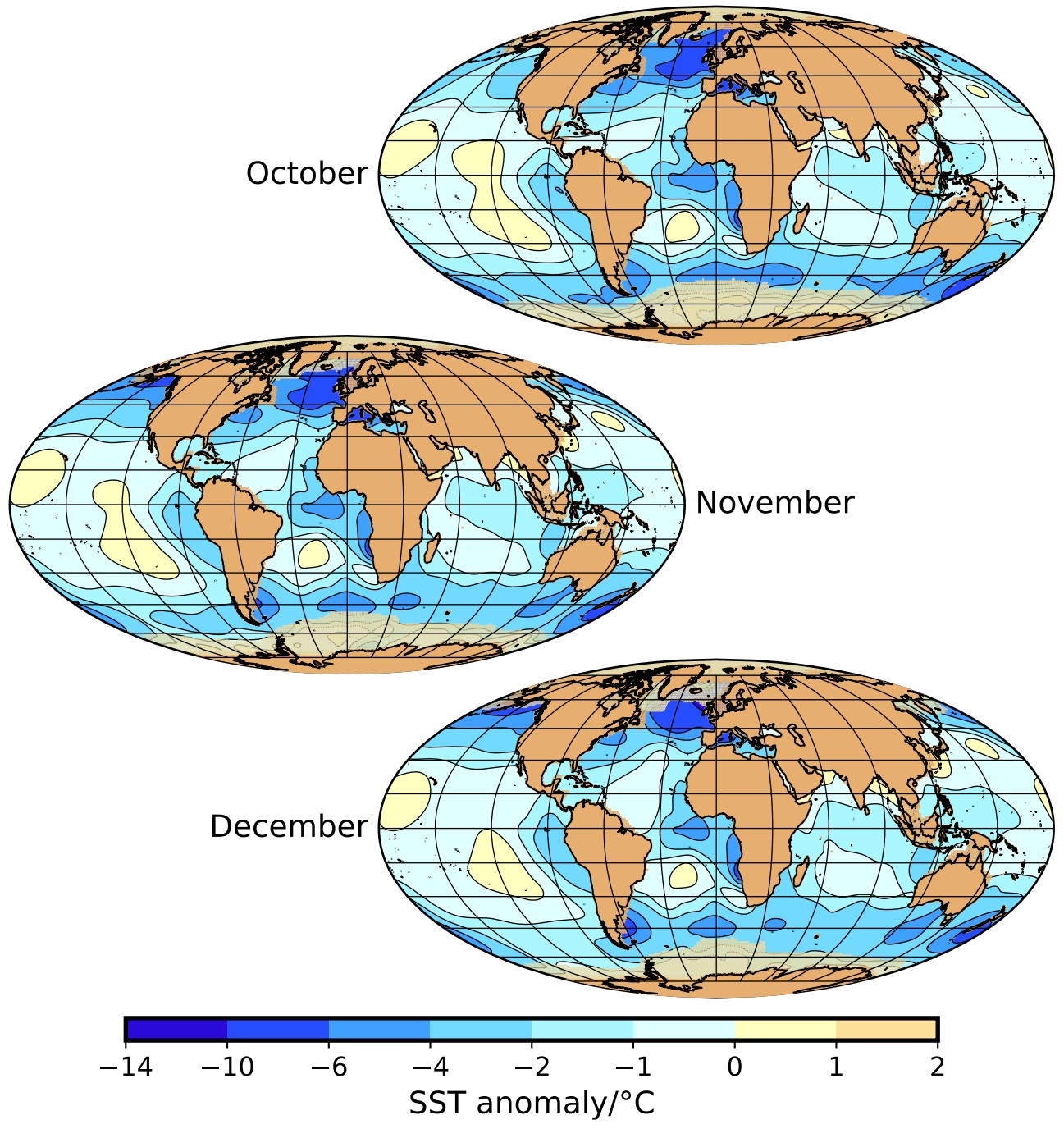

**Figure A8.** Sea-surface temperature anomaly (contour map) and sea-ice extent (yellow-brownish areas close to Antarctica and in the Arctic) for October, November and December

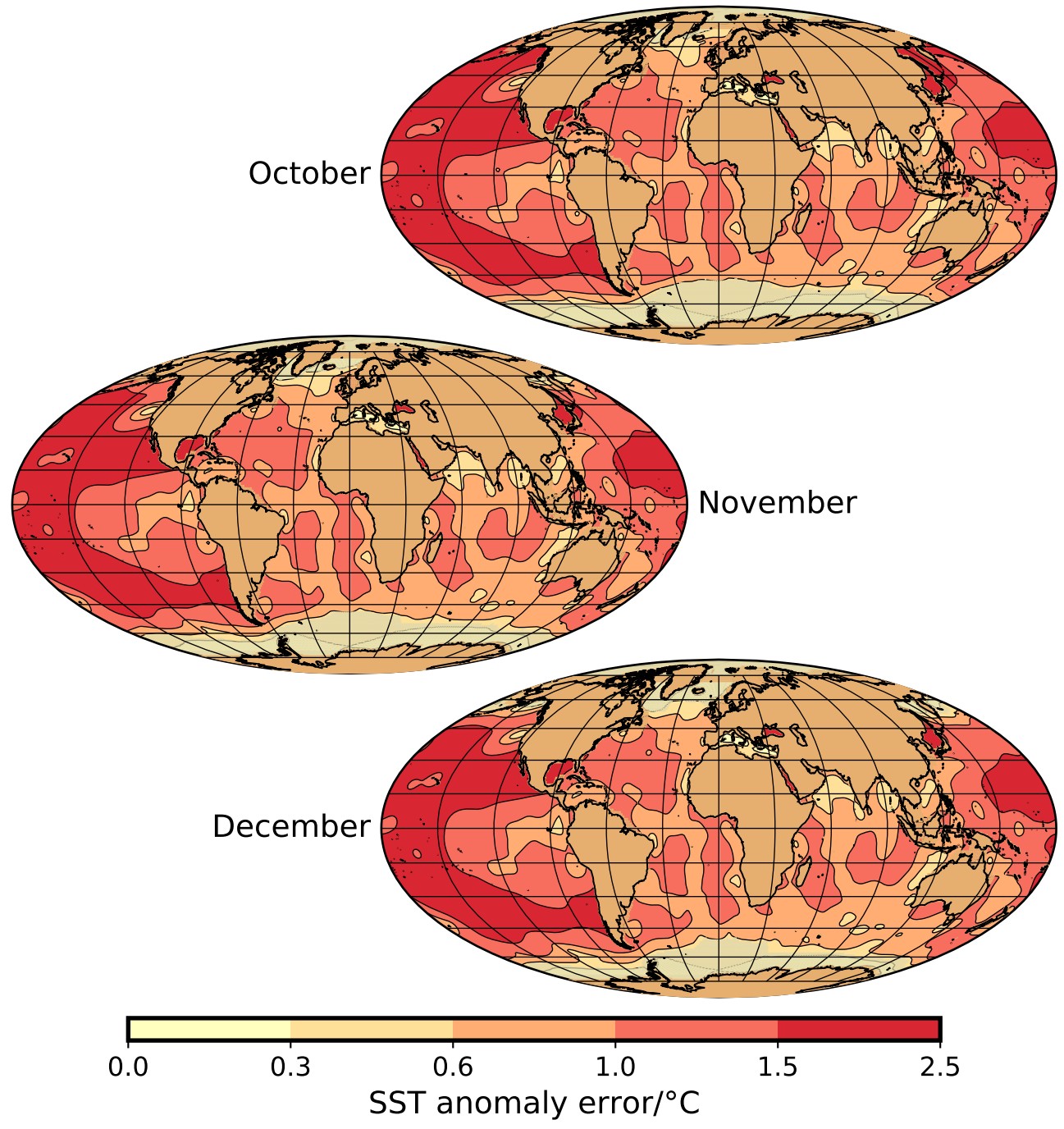

**Figure A9.** Uncertainty of NSST anomaly (contour map) and sea-ice extent (yellow-brownish areas close to Antarctica and in the Arctic, cf. Fig. A8) for October, November and December