# Peer review of "A global climatology of the ocean surface during the Last Glacial Maximum mapped on a regular grid (GLOMAP)"

_Climate of the Past, 2019_

## Referee Comment (RC1) · Anonymous Referee #1 · 3 Apr 2020

The paper presents **GLOMAP**, a new climatology for SST and sea-ice extension for a specific period and which covers the World Ocean. According to the journal guideline, my overview is as follows:

**Scientific significance:** the manuscript represents a substantial contribution in the terms of data (i.e. a new climatology) and application of a method (DIVA) to a specific type of data. The final products (climatology) will certainly be useful to other scientists and employed in different contexts.

**Scientific quality:** the scientific approach and applied methods are valid: the data pro-

cessing is well described, the limitations and the uncertainties on the data are clearly presented.

**Presentation quality:** the manuscript and the figures are clear, the document is concise and well structured.

**General comments**

The authors state is that the *Data-Interpolating Variational Analysis* method is also capable of analyzing much sparser data. In the paper there is no real comparison or assessment of other methods that would allow the author to make this statement. So I would encourage the authors to add a few lines explaining their choice, maybe by adding some details on the methods that could justify their choice for the present study would be relevant and comparing with the methods employed to create **CLIMAP** and **GLAMAP** climatologies.

Data availability: it would be useful for the reader to have the direct URL to avoid searching within the PANGEA database. I searched using "GLOMAP" as keyword (https://www.pangaea.de/?q=GLOMAP) but that request did not return any result, so I guess the data will be published once the paper is published. What is the format of the climatology?

Some parts of the processing workflow (Section 2) were not totally clear to me, for example
lines 89-109: why are the two steps necessary, and why not use all the data at the same time for the monthly interpolations?
lines 94-94: To each sea-ice covered data point we assigned an error of 2°C
$\rightarrow$ does this mean that measurements taken where it is supposed to be sea ice are used for the gridding? From line 80, it seems that the finite-element mesh is based on

a coastline from a glacial topography, so the measurements on ice would not influence the analysis.

Could you add the figure of the coastline and the finite-element mesh in the Appendix?

**Minor comments and typos**

39: This method allows to take → allows one to take

39: the uncertainty on the reconstruction → the authors probably means the uncertainties on the observations (instrumental errors, representativeness errors etc).

40: or for assessing the data-analysis mismatch.
→ independently of the interpolation technique, the data-analysis mismatch is not always a relevant metrics: one can obtain a very small mismatch by forcing the analysis to be close to the observation. This would result in a "noisy" or "patchy" interpolated field, which may not represent what a climatology should be.

line 49: "we digitized sea-ice edges"
→ can you explicit what is the process to digitize? For Xiao et al. (2015, Fig. 7a), the panel a of their figure did not display any coordinates, how was that solved? Does this also means that no other publication provides the sea-ice edges in digital format?

65 is associate with → associated with

72-73 the magnitude of the data (anomalies) themselves as well as on the gradients, the variability and data-analysis misfits

82 We fitted the covariance function to the foraminiferal data
→ indicate how many data points were considered for the fit.

84 estimates of the correlation length of 9.2° and 10.2°

→ is there a physical explanation to this difference, or do you attribute that to numerics?

94 To each sea-ice covered data point we assigned an error of 2 °C
→ is it necessary, since you defined a coastline and mask using "glacial topography GLAC-1D"

103: new (artificial) diatom and radiolaria data
→ what is the source of these data? (and why "artificial"?)

141 South Alantic → South Atlantic

169-170 DIVA may be used to more accurately estimate the spatial covariances as described by the non-diagonal terms
→ Beckers et al. (2014) may be relevant for this aspect

Beckers, J.-M.; Barth, A.; Troupin, C. & Alvera-Azcárate, A. Some approximate and efficient methods to assess error fields in spatial gridding with DIVA (Data Interpolating Variational Analysis) (2014). *Journal of Atmospheric and Oceanic Technology*, **31:** 515-530. doi:10.1175/JTECH-D-13-00130.1

224 may allow to first smooth
→ may allow one to first smooth

Figure 2:

- indicate the meaning of the yellow-brownish area close to the Antarctica (rectangles in the attached figure)

- Analyzed SST anomalies
  → with respect to what reference or background are computed the anomalies? (also in text, line 97).

- the size of the dots representing the data is a little bit to large, so several dots overlap, especially in the northern part of the domain. In Figure 4 the dots are

smaller.

334 WOA: World Ocean Atlas 1998 → why not use a more recent version of the World Ocean Atlas?

Figures for the monthly fields: having 6 (or maybe 12) sub-figures seems possible and won't cause problem to the readability of the plots.

———————————————————

[Figure]

**Fig. 1.**

---

## Referee Comment (RC2) · Jessica Tierney (Referee) · 20 Apr 2020

General comments:

Paul et al. present a spatial reconstruction of LGM SST anomalies and sea-ice extent using a new interpolation method, DIVA. Since the underlying data is mostly the MARGO data, the authors get generally the same results as MARGO (2009), with perhaps even smaller anomalies for glacial cooling. I have a couple of general comments here about the data and method used:

1) Underlying data. The sea-ice reconstruction appears to be based on both faunal

assemblages and biomarker evidence like IP25, but the SST reconstruction is based only on faunal data from the MARGO collection. Why is this? I'm not sure why the authors would not use the geochemical data in the MARGO collection (?) If there is a reason, then it should be made clear. There is arguably value in a single-proxy field reconstruction, but it should be justified. Also, I think some of the faunal data may have no-analog issues. Were these dealt with in any way?

A bigger problem with the choice of data is that MARGO is over a decade old now, and surely there have been more faunal datasets published since then (Certainly, there is far more geochemical data available now). Since the authors just use the MARGO data, they get results that nearly the same as MARGO. This doesn't seem like an advance in our understanding of the LGM. If the purpose of this paper is provide new insights into the LGM, I would suggest that the authors consider updating their dataset. If the purpose of this paper is to demonstrate a method (DIVA) then the cooling and ECS results should be downplayed.

2) DIVA method. This method is new to me, but seems appropriate for the problem at hand. However some more description of the method is needed here for non-specialists. I'm also wondering, given that DIVA was designed to work with more dense modern oceanographic data, how well it does with the sparse data of the LGM? Can the authors do some validation tests to assess this? E.g., withhold 10-25% of the data, fit the field using DIVA, then validate on the withheld data? This would provide some sense of performance.

3) Comparisons to other field reconstructions of the LGM. The authors discuss how their result is fairly similar to MARGO, which is not surprising since the underlying data are similar. What about other products? There are some data assimilation products to compare with (Annan and Hargreaves, 2013) - Paul is a co-author on one of them (Kurahaski-Nakamura et al., 2017 https://doi.org/10.1002/2016PA003001) and see also Amrhein et al., 2018 (https://doi.org/10.1175/JCLI-D-17-0769.1). We have a new data assimilation product available as well (in review, but a preprint is available

here: https://eartharxiv.org/me5uj/) based on an updated database of geochemical proxies.

4) Estimates of glacial cooling and climate sensitivity. In keeping with the MARGO results, these are arguably unrealistically low (global SST change of -1.7, ECS of 1.5). The MARGO-based results of ECS (Schmittner et al. 2011) have faced a lot of criticism. Multiple studies have suggested a global SST change closer to -3C (Ballantyne et al., 2005, Lea et al., 2000) and a corresponding global air temperature change closer to 5-6C (e.g. Snyder, 2016, Nature; Schneider von Deimling et al., 2006 GRL, Holden et al., 2010 Climate Dynamics, Bereiter et al., 2018, Nature, and our new estimate in our preprint noted above). There needs to be a critical discussion in light of these other results.

Also: how was ECS calculated? There must be a scaling assumption to translate to global mean surface temperature, and then there has to be estimation of the forcing as well (the denominator). Please describe this.

My overall take of this paper is: It's really interesting to see the application of DIVA to paleoclimate information, and this could use some more discussion and exploration (perhaps comparison to optimal interpolation). However in terms of providing new scientific insights into the LGM, the paper is limited here by use of the MARGO faunal dataset, which ultimately shapes the results. No matter what the method used, the MARGO data, particularly the assemblage data, provide an estimate of glacial cooling that is very small. This result has been challenged a lot over the years and there is a sense that perhaps no-analog problems still plague the faunal data.

I think the best solution here would be to update the underlying dataset with new studies - either new faunal data or new faunal data + geochemical data. Otherwise, the conclusions of the paper re: glacial cooling and climate sensitivity are just the same as MARGO.

Alternatively, the authors could treat this paper as a methods paper. If the goal is to

just demonstrate application of DIVA, then it's OK to stick with MARGO. But in that case comparisons should be made with other field estimation methods (OI, data assimilation) and the scientific results (LGM cooling and ECS) should be downplayed and presented critically since they are ultimately tied to the underlying data.

Specific comments:

Abstract: Clarify that GLOMAP is based on only faunal transfer function data (except for the use of IP25 for sea ice).

Section 2.1: Please clarify here what each reconstruction is based on (transfer functions, IP25, etc).

Section 2.2: Why did you only use the faunal data from the MARGO collection? Also MARGO is now 11 years old. I imagine that more data have been published since then. Certainly for the geochemical proxies this is true. I think it's worth updating the data with newly published results.

Section 2.3: This section could benefit from a little more explanation of how DIVA works since most readers will not be familiar with Troupin et al. (2012). In particular, it would be useful to describe how DIVA is distinct from pure interpolation (no information about spatial relationships) vs. optimal interpolation (covariance structure is set). Is DIVA essentially isotropic away from the coastlines?

Section 3.1: The use of past tense here is a little confusing. Use present tense for describing the results.

---

## Author Comment (AC1) · 9 Jun 2020

**A global climatology of the ocean surface during the Last Glacial Maximum mapped on a regular grid (GLOMAP) – Final response to referee comments**

André Paul[1], Stefan Mulitza[1], Ruediger Stein[1, 2], and Martin Werner[2]

[1] MARUM – Center for Marine Environmental Sciences and Department of Geosciences, University of Bremen, Bremen, Germany
[2] Alfred Wegener Institute, Helmholtz Centre for Polar and Marine Research (AWI), Bremerhaven, Germany

**Response to Anonymous Referee #1**

We are thankful to the referee for the helpful and constructive comments that will surely improve our manuscript.

**General comments**

*The authors state is that the Data-Interpolating Variational Analysis method is also capable of analyzing much sparser data.*
*In the paper there is no real comparison or assessment of other methods that would allow the author to make this statement.*
*So I would encourage the authors to add a few lines explaining their choice, maybe by adding some details on the methods that*
*could justify their choice for the present study would be relevant and comparing with the methods employed to create CLIMAP*
*and GLAMAP climatologies.*

- – In response to the valid concern raised by both referees, in the revised manuscript we will provide a test of the method by, as proposed by the second referee, withholding a certain fraction of the data, make a fit to the remaining data using DIVA, then compare the fit to the withheld data.

- – Furthermore, we will take the opportunity to explain our choice of using DIVA – for example, DIVA takes the coastlines into account, since an underlying variational principle is solved only on a finite-element mesh that covers the sea. This prevents the exchange of information across boundaries such as land bridges, peninsulas or islands, which might produce artificial mixing between, for example, Pacific and Atlantic water masses across the Panama isthmus. In solving the variational principle, it not only takes into account the distance between analysis and data, but also imposes a smoothness constraint and, if desired, an advection constraint, and moreover, it provides an uncertainty estimate.

- – Regarding the method employed to create the CLIMAP climatology: As described in detail by Broccoli and Marciniak (1996; see also Manabe and Broccoli, 2020), CLIMAP used a subjective analysis procedure (i.e. contouring by hand) to yield the paleoisotherm maps (CLIMAP Project Members, 1976, 1981), which were then digitized on a regular grid.

- With respect to the GLAMAP climatology, different methods were applied: Contouring of the paleotemperature maps was also by hand, and the isotherms were derived by means of visual triangulation from strictly linear interpolation between the SST reconstructions at the irregularly distributed neighbor sites (Sarnthein et al., 2003; Pflaumann et al., 2003). For gridding, either the digitized isotherms (Paul and Schäfer-Neth, 2003) or the SST reconstructions at the sediment core positions (Schäfer-Neth and Paul, 2004) were objectively interpolated using variogram analysis and kriging in spherical coordinates; and the resulting gridded fields were compared (Schäfer-Neth and Paul, 2004, Fig. 5). The seasonal cycle was constructed in the same way as for the GLOMAP climatology: Following the PMIP (1993) guidelines, a sinusoidal cycle was fitted to the glacial-to-modern anomalies and then the modern monthly SST (taken as 10 m data from the WOA, 1998) was added.

- The variogram analysis and kriging cannot deal easily with coastlines, for example, it may take into account data points separated by a land bridge or an island. This was one motivation to apply the DIVA method (Troupin et al., 2012), which employs a finite-element mesh derived from a given topography.

*Data availability: it would be useful for the reader to have the direct URL to avoid searching within the PANGEA database. I searched using "GLOMAP" as keyword (https://www.pangaea.de/?q=GLOMAP) but that request did not return any result, so I guess the data will be published once the paper is published. What is the format of the climatology?*

Indeed, the final version of the data will be published in the PANGEA database once the paper is published. The format of the climatology is Network Common Data Format (netCDF).

*Some parts of the processing workflow (Section 2) were not totally clear to me, for example lines 89-109: why are the two steps necessary, and why not use all the data at the same time for the monthly interpolations?*

We took two steps in order to make use of the diatom and radiolarian data from the Southern Ocean. Because in this region the biogenic particle flux to the sea floor is restricted to austral summer, even in areas unaffected by sea-ice cover (Abelmann and Gersonde, 1991; Gersonde and Zielinski, 2000; Fischer et al., 2002), only Southern Hemisphere summer (JFM) SST has been estimated by Gersonde et al. (2005). Therefore, in the first step, we only used the foraminiferal and dinoflagellate data for JAS and JFM. In the second step, we included the diatom and radiolarian data available for JFM and filled in the missing data for JAS by taking the results from the first step at the grid points where diatom and radiolaria data for JFM exist. In this way we were able to create monthly data at all grid points where data exist and repeat the DIVA analysis.

*lines 94-94: To each sea-ice covered data point we assigned an error of $2°C$ $\longrightarrow$ does this mean that measurements taken where it is supposed to be sea ice are used for the gridding? From line 80, it seems that the finite-element mesh is based on a coastline from a glacial topography, so the measurements on ice would not influence the analysis.*

Indeed, we used a glacial topography to generate the coastlines, however, these coastlines do not reflect the sea-ice edges. Therefore the finite-element mesh extended to the sea-ice covered regions. To utilze the information on past sea-ice coverage, we digitized the sea-ice edges, created monthly sea-ice masks and included the sea-ice covered data points in the DIVA analysis. A relatively large error of 2 °C was assigned to each sea-ice covered data point to reflect the uncertainty in the LGM sea-ice extent reconstructions.

55 *Could you add the figure of the coastline and the finite-element mesh in the Appendix?*

 Yes, we can certainly do that, thank you for this suggestion!

**Minor comments and typos**

 *39: This method allows to take* ⟶ *allows one to take*

 Corrected

60 *39: the uncertainty on the reconstruction* ⟶ *the authors probably means the uncertainties on the observations (instrumental errors, representativeness errors etc).*

 Indeed, we mean the paleo-data. Because these are not direct observations as instrumental data, but indirect estimates derived from fossil faunal assemblages using a transfer function technique, we often use the term "reconstruction", which may be confusing. Hence in the sentence in question we will replace "reconstruction" by "paleo-data".

65 *40: or for assessing the data-analysis mismatch.* ⟶ *independently of the interpolation technique, the data-analysis mismatch is not always a relevant metrics: one can obtain a very small mismatch by forcing the analysis to be close to the observation. This would result in a "noisy" or "patchy" interpolated field, which may not represent what a climatology should be.*

 This is certainly true, thank you for pointing this out. Figures 3 and 4 show that the interpolated fields are neither "noisy" or
70 "patchy", something we will take up in the discussion of our results.

 *line 49: "we digitized sea-ice edges"* ⟶ *can you explicit what is the process to digitize? For Xiao et al. (2015, Fig. 7a), the panel a of their figure did not display any coordinates, how was that solved? Does this also means that no other publication provides the sea-ice edges in digital format?*

 It is indeed true that none of the publications provides the sea-ice edges in digital format. Therefore we had to digitize the
75 curves from the published maps to obtain their location in geographic coordinates. In case of Xiao et al. (2015, Fig. 7a), neither the projection nor the coordinates are given, hence we used the few indicated topographic features (islands) and the sediment core locations to take into account a summer ice edge north of the Barents Sea in our sea-ice mask.

 *65 is associate with* ⟶ *associated with*

 Corrected

80 *72-73 the magnitude of the data (anomalies) themselves as well as on the gradients, the variability and data-analysis misfits*

 Rewritten as follows: the magnitude of the data (anomalies) themselves as well as on the gradients, the variability and data-analysis misfits

 *82 We fitted the covariance function to the foraminiferal data* ⟶ *indicate how many data points were considered for the fit.*

 There were 444 data points considered for the fit. We will add this number to the revised manuscript.

85 *84 estimates of the correlation length of 9.2° and 10.2°* ⟶ *is there a physical explanation to this difference, or do you attribute that to numerics?*

 We attribute this difference to more noise and a longer tail of the data covariance for the July-August-September (JAS) season that results in a slightly larger correlation length of the fitted covariance.

*94 To each sea-ice covered data point we assigned an error of 2 °C → is it necessary, since you defined a coastline and mask using "glacial topography GLAC-1D"*

Please see our response to your general comment on lines 94–94.

*103: new (artificial) diatom and radiolaria data → what is the source of these data? (and why "artificial"?)*

The diatom and radiolarian data for JAS are "artificial" in the sense that they were generated from the results of the first step of the DIVA analysis at the grid points where diatom and radiolaria data for JFM exist (please see our response to your general comment on lines 89–109).

*141 South Alantic → South Atlantic*

Corrected

*169-170 DIVA may be used to more accurately estimate the spatial covariances as described by the non-diagonal terms → Beckers et al. (2014) may be relevant for this aspect Beckers, J.-M.; Barth, A.; Troupin, C. & Alvera-Azcárate, A. Some approximate and efficient methods to assess error fields in spatial gridding with DIVA (Data Interpolating Variational Analysis) (2014). Journal of Atmospheric and Oceanic Technology, 31: 515-530. doi:10.1175/JTECH-D-13-00130.1*

Thank you for pointing out this reference to us. We will refer to it in the revised manuscript.

*224 may allow to first smooth → may allow one to first smooth*

Corrected

*Figure 2:*

- *indicate the meaning of the yellow-brownish area close to the Antarctica (rectangles in the attached figure)*

We apologize for the missing information in the figure caption. The yellow-brownish areas close to Antarctica and in the Arctic ocean indicate the LGM sea-ice masks based on the selected LGM sea-ice reconstructions.

- *Analyzed SST anomalies → with respect to what reference or background are computed the anomalies? (also in text, line 97).*

The LGM estimates from the MARGO database are themselves anomalies that are calculated with respect to World Ocean Atlas 1998. We will clarify this in the Methods section of the revised manuscript.

- *the size of the dots representing the data is a little bit to large, so several dots overlap, especially in the northern part of the domain. In Figure 4 the dots are smaller.*

Thank you for pointing this out to us. We will reduce the size of the dots representing the data and use the same size in all figures.

*334 WOA: World Ocean Atlas 1998 → why not use a more recent version of the World Ocean Atlas?*

The main reason for using the 1998 version of the World Ocean Atlas is to be consistent with the MARGO database, because it is this version that was used for calibrating the methods which were used to reconstruct the LGM estimates and served as a reference for the LGM SST anomalies. Furthermore, the majority of the data that entered the 1998 version represents

observations that were taken before the intensification of global warming towards the end of the 20$^{\text{th}}$ century, thereby reducing the anthropogenic imprint.

*Figures for the monthly fields: having 6 (or maybe 12) sub-figures seems possible and won't cause problem to the readability of the plots.*

125    Thank you fo this suggestion. We will try combining 6 or even 12 sub-plots in one figure and check its readability, especially of the dots representing the data.

**References**

Abelmann, A. and Gersonde, R.: Biosiliceous particle flux in the Southern Ocean, Marine Chemistry, 35, 503–536, 1991.

Broccoli, A. J. and Marciniak, E. P.: Comparing simulated glacial climate and paleodata: A reexamination, Paleoceanography, 11, 3–14, 1996.

CLIMAP Project Members: The Surface of the Ice-Age Earth, Science, 191, 1131, https://doi.org/10.1126/science.191.4232.1131, 1976.

CLIMAP Project Members: Seasonal reconstructions of the Earth's surface at the Last Glacial Maximum, Geological Society of America, Map and Chart Series, MC-36, 1–18, 1981.

Fischer, G., Gersonde, R., and Wefer, G.: Organic carbon, biogenic silica and diatom fluxes in the marginal winter sea ice zone and in the Polar Front Region in the Southern Ocean (Atlantic sector): interannual variation and changes in composition, Deep-Sea Research II, 49, 1721–1745, https://doi.org/10.1016/S0967-0645(02)00009-7, 2002.

Gersonde, R. and Zielinski, U.: The reconstruction of late Quaternary Antarctic sea-ice distribution –the use of diatoms as proxies for sea-ice, Paleogeography, Paleoclimatology and Paleoecology, 162, 263–286, https://doi.org/10.1016/S0031-0182(00)00131-0, 2000.

Gersonde, R., Crosta, X., Abelmann, A., and Armand, L.: Sea-surface temperature and sea ice distribution of the Southern Ocean at the EPILOG Last Glacial Maximum – a circum-Antarctic view based on siliceous microfossil records, Quaternary Science Reviews, 24, 869–896, https://doi.org/doi:10.1016/j.quascirev.2004.07.015, 2005.

Manabe, S. and Broccoli, A. J.: Beyond Global Warming, Princeton University Press, https://doi.org/10.1515/9780691185163, 2020.

Paul, A. and Schäfer-Neth, C.: Modeling the water masses of the Atlantic Ocean at the Last Glacial Maximum, Paleoceanography, 18, doi:10.1029/2002PA000 783, 2003.

Pflaumann, U., Sarnthein, M., Chapman, M., Funnel, B., Huels, M., Kiefer, T., Maslin, M., Schulz, H., Swallow, J., van Kreveld, S., Vautravers, M., Vogelsang, E., and Weinelt, M.: The Glacial North Atlantic: Sea-surface conditions reconstructed by GLAMAP-2000, Paleoceanography, 18, doi:10.1029/2002PA00 774, 2003.

PMIP: Paleoclimate Modelling Intercomparison Project, http://www-pcmdi.llnl.gov/pmip/newsletters/newsletter02.html, Tech. rep., 1993.

Sarnthein, M., Gersonde, R., Niebler, S., Pflaumann, U., Spielhagen, R., Thiede, J., Wefer, G., and Weinelt, M.: Overview of Glacial Atlantic Ocean Mapping (GLAMAP 2000), Paleoceanography, 18, doi:10.1029/2002PA00 769, 2003.

Schäfer-Neth, C. and Paul, A.: The Atlantic Ocean at the Last Glacial Maximum: 1. Objective mapping of the GLAMAP sea-surface conditions, in: The South Atlantic in the Late Quaternary: Reconstruction of Material Budgets and Current Systems, edited by Wefer, G., Mulitza, S., and Ratmeyer, V., pp. 531–548, Springer-Verlag, Berlin, Heidelberg, 2004.

Troupin, C., Barth, A., Sirjacobs, D., Ouberdous, M., Brankart, J.-M., Brasseur, P., Rixen, M., Alvera-Azcárate, A., Belounis, M., Capet, A., Lenartz, F., Toussaint, M.-E., and Beckers, J.-M.: Generation of analysis and consistent error fields using the Data Interpolating Variational Analysis (DIVA), Ocean Modelling, 52-53, 90–101, https://doi.org/10.1016/j.ocemod.2012.05.002, http://modb.oce.ulg.ac.be, 2012.

WOA: World Ocean Atlas 1998, Tech. rep., National Oceanographic Data Center, Silver Spring, Maryland, 1998.

Xiao, X., Stein, R., and Fahl, K.: MIS 3 to MIS 1 temporal and LGM spatial variability in Arctic Ocean sea ice cover: Reconstruction from biomarkers, Paleoceanography, 30, 969–983, https://doi.org/10.1002/2015PA002814, 2015.

---

## Author Comment (AC2) · 9 Jun 2020

**A global climatology of the ocean surface during the Last Glacial Maximum mapped on a regular grid (GLOMAP) – Final response to referee comments**

André Paul[1], Stefan Mulitza[1], Ruediger Stein[1, 2], and Martin Werner[2]

[1] MARUM – Center for Marine Environmental Sciences and Department of Geosciences, University of Bremen, Bremen, Germany
[2] Alfred Wegener Institute, Helmholtz Centre for Polar and Marine Research (AWI), Bremerhaven, Germany

**Response to Referee #2 (Jessica Tierney)**

We are thankful to the referee for her insightful comments which we feel will substantially improve our manuscript.

**General comments**

*Paul et al. present a spatial reconstruction of LGM SST anomalies and sea-ice ex- tent using a new interpolation method, DIVA.*
*Since the underlying data is mostly the MARGO data, the authors get generally the same results as MARGO (2009), with per-*
*haps even smaller anomalies for glacial cooling. I have a couple of general comments here about the data and method used:*

*1) Underlying data. The sea-ice reconstruction appears to be based on both faunal assemblages and biomarker evidence*
*like IP25, but the SST reconstruction is based only on faunal data from the MARGO collection. Why is this? I'm not sure why*
*the authors would not use the geochemical data in the MARGO collection (?) If there is a reason, then it should be made clear.*
*There is arguably value in a single-proxy field reconstruction, but it should be justified. Also, I think some of the faunal data*
*may have no-analog issues. Were these dealt with in any way?*

We agree that we should make clear why we base the gridded SST on the faunal data only. In fact, there are a number of reasons:

- Our interest is in a monthly climatology of the SST anomaly and sea-ice extent during the LGM, and while faunal
assemblages are not without issues, they are the only sedimentary proxy that can provide a seasonal reconstruction.

- In addition, a single-proxy reconstruction has the advantage of internal consistency between the different sediment core sites, thereby reducing the noise.

- As for the MARGO reconstruction of the northern North Atlantic Ocean, all four proxies (planktonic foraminifer assemblages, dinocyst assemblages, alkenone coccolithophorid biomarkers and Mg/Ca ratios in planktonic forminifers)
support the same features of sea-ice cover (de Vernal et al., 2006; see also Sarnthein et al., 2003). The IP25/PIP25 data by Xiao et al. (2015, Fig. 7a) and Méheust et al. (2018) add information for the Barents Sea and the North Pacific Ocean, respectively. However, in the Nordic Seas, there are large discrepancies between the different SST reconstructions, well above their level of uncertainties (de Vernal et al., 2006). We suspect that the apparently warm signal recorded by dinocyst assemblages, coccolithophores and alkenones may be due long-distance lateral transport (de Vernal et al., 2006; Rühlemann and Butzin, 2006), whereas foraminifera-based proxies have the advantage that they drop very quickly to the sediment (Takahashi and Be, 1984). Further possible sources of errors are the overwintering of dinoflagellates in a cyst phase as well as their broad tolerances for temperature (Dale, 2001). Using alkenones for SST reconstructions might be problematic in high latitudes because of the low sensitivity of the calibration of alkenones at low temperature (Conte et al., 2006), the possibility of redeposition of old and warm signal carrying alkenones with particulate matter originating from the glaciated continental margins and once more the influence of alkenones transported by currents from warmer areas into the polar regions (e.g., Bendle and Rosell-Melé, 2004; Filippova et al., 2016).

– Furthermore, foraminiferal assemblages are usually dominated by species adapted to the environment of the overlying water column (Morey et al., 2005). We therefore consider the temperature estimation to be more robust against the expatriation of single shells that can affect proxies measured on monospecific samples.

– Finally, proxies based on the chemistry of shells of living organisms suffer from the inherent problem that the environmental sensitivity of that organism biases the recording of the proxy (Mix, 1987; Fraile et al., 2009). The transfer function method does not have this problem since it actually uses the environmental sensitivity of foraminifera.

The MARGO project carefully dealt with the no-analog problem in a number of ways. For example, Gersonde et al. (2005) discard all samples with no analogs (dissimilarity > 0.25) and when the majority of the samples in the LGM interval has no anlogs, the estimated quality level is downgraded to 3. Kucera et al. (2005) combine three methods (Artificial Neural Networks – ANN, Revised Analog Method – RAM, Maximum Similarity Technique – SIMMAX) in a multi-technique approach that facilitates a test of the robustness of SST estimates and provides a means to identify potential no-analog conditions or faunas.

*A bigger problem with the choice of data is that MARGO is over a decade old now, and surely there have been more faunal datasets published since then (Certainly, there is far more geochemical data available now). Since the authors just use the MARGO data, they get results that nearly the same as MARGO. This doesn't seem like an advance in our understanding of the LGM. If the purpose of this paper is provide new insights into the LGM, I would suggest that the authors consider updating their dataset. If the purpose of this paper is to demonstrate a method (DIVA) then the cooling and ECS results should be downplayed.*

An important purpose of our manuscript is indeed to demonstrate the applicability of DIVA to sparse and irregularly spaced paleoceanographic data. Although an update of MARGO is certainly important, it is not clear whether this would change the results significantly, and it is beyond the scope of this manuscript.

We agree that we cannot expect to find truly new results and that we need to make that clearer in the revised manuscript, but we actually see a number of advantages using the well-established MARGO database: To date, it still provides the most comprehensive dataset of LGM SST anomalies at the sediment core locations including their error estimates. Without creating a spatially complete, gridded climatology the MARGO project has not been able to provide area- and uncertainty-weighted global and regional averages or boundary conditions. The main motivation of our manuscript is to provide a spatially complete, continuous gridded climatology for such purposes. By using MARGO, we can directly compare the gridded and non-gridded data and assess the effect of proper weighting in calculating global and regional averages, and we indeed find small but noticeable differences.

Another advantage of the MARGO database is the consistent use of the modern SST (WOA, 1998) as the calibration as well as the reference dataset. Most methods used for temperature reconstruction are calibrated to absolute temperatures, not temperature differences or changes. By using the calibration dataset as a reference dataset, any error is only incurred once in calculating the LGM anomaly. If using any other time slice (e.g., the Late Holocene) as a reference dataset, the error would double.

With respect to downplaying the cooling and ECS results, please see our response to the major point no. 4.

*2) DIVA method. This method is new to me, but seems appropriate for the problem at hand. However some more description of the method is needed here for non-specialists. I'm also wondering, given that DIVA was designed to work with more dense modern oceanographic data, how well it does with the sparse data of the LGM? Can the authors do some validation tests to assess this? E.g., withhold 10-25% of the data, fit the field using DIVA, then validate on the withheld data? This would provide*
*some sense of performance.*

Thank you for suggesting this useful test, which we will provide in the revised manuscript to give an idea of the performance, together with some more explanation of the method for non-specialists.

*3) Comparisons to other field reconstructions of the LGM. The authors discuss how their result is fairly similar to MARGO, which is not surprising since the underlying data are similar. What about other products? There are some data assimilation*
*prod- ucts to compare with (Annan and Hargreaves, 2013) - Paul is a co-author on one of them (Kurahaski-Nakamura et al., 2017 https://doi.org/10.1002/2016PA003001) and see also Amrhein et al., 2018 (https://doi.org/10.1175/JCLI-D-17-0769.1). We have a new data assimilation product available as well (in review, but a preprint is available here: https://eartharxiv.org/me5uj/) based on an updated database of geochemical proxies.*

Thank you for this valuable suggestion, too. In the revised manuscript, we will follow up on it and compare our gridded
climatology with other products and discuss the advantages and disadvantages of the various methods. For example, while DIVA is a purely statistical method, the adjoint method used by Kurahashi-Nakamura et al. (2017) and Amrhein et al. (2018) makes use of the physics and paramaterizations of an ocean general circulation model (MITgcm), however, it also inherits the biases of this model (similar to the enhanced cooling in the eastern equatorial Pacific that reflects the CESM prior in Tierney et al., 2019). In contrast, Annan and Hargreaves (2013) employ an ensemble of models, but with partly inconsistent physics
and parameterizations.

*4) Estimates of glacial cooling and climate sensitivity. In keeping with the MARGO results, these are arguably unrealistically low (global SST change of -1.7, ECS of 1.5). The MARGO-based results of ECS (Schmittner et al. 2011) have faced a lot of criticism. Multiple studies have suggested a global SST change closer to -3C (Ballantyne et al., 2005, Lea et al., 2000) and a corresponding global air temperature change closer to 5-6C (e.g. Snyder, 2016, Nature; Schneider von Deimling et al., 2006*

*GRL, Holden et al., 2010 Climate Dynamics, Bereiter et al., 2018, Nature, and our new estimate in our preprint noted above).*
*There needs to be a critical discussion in light of these other results.*

According to Bindoff et al. (2013), the equilibrium climate sensitivity is likely in the range of 1.5 °C to 4.5 °C ("high confidence"), extremely unlikely less than 1°C ("high confidence"), and very unlikely greater than 6°C ("medium confidence"). The PALAEOSENS Project Members (2012) estimate based on a comprehensive set of reconstructions of past temperatures and radiative forcing yields a range of 2.2 °C to 4.8°C for the actual climate sensitivity. While neither a value of the equilibrium climate sensitivity at the low end (about 1.5 °C) nor at the high end (about 4.5 °C) can be ruled out for purely scientific reasons, we agree that a critical discussion of our result in the light of these and other results is needed. In fact, it is our area- and uncertainty-weighted gridded climatology that enables us to reassess the MARGO results.

The study by Schmittner et al. (2011) was mainly criticized because of some methodological issues (e.g., missing atmo- spheric feedbacks, a definition of climate sensitivity that includes feedbacks associated with vegetation, an insufficiently steep lapse rate, underprediction of land temperatures and misfit of ocean data in mid- and high latitudes – see the discussion in RealClimate, 2011). These resulted in a global cooling during the LGM in the range of 1.7 °C to 3.7 °C, which appears to be too small and possibly led to an underestimate of the climate sensitivity with a surprisingly low uncertainty range of 1.4 °C to 2.8 °C. Indeed, Annan and Hargreaves (2013), who also used MARGO as ocean data, arrived at a larger global cooling during the LGM of 3.1 °C to 4.7 °C, supposedly implying a larger climate sensitivity.

The best resolved alkenone-based SST estimates from the central Pacific show an SST change between 1.2 °C and 2 °C (Prahl et al., 1989; Lee et al., 2001; de Garidel-Thoron et al., 2007). From a number of studies using Mg/Ca as well as alkenones, Lea (2004) finds a tropical cooling at the LGM by 2.8 °C ± 0.7 °C. Leduc et al. (2017) summarize the results of the Sensitivity of the Tropics (SENSETROP) working group, which after the incorporation of high-quality records and a thorough quality control obtains a cooling of the low latitudes during the LGM of -2.3 °C ± 0.8 °C and -2.4 °C ± 0.8°C for alkenone- and Mg/Ca-based SST estimates, respectively. Tierney et al. (2019) obtain a very similar mean tropical cooling of -2.5 °C (-2.8 °C to -2.2 °C, 95 % CI) from the SST proxies on their own. These values are indeed larger than the estimates by CLIMAP, MARGO and Annan and Hargreaves (2013) by up to a degree, but not as large as the early estimate from corals by Guilderson et al. (1994) of about 5 °C, which made Crowley (2000) wonder whether it would be possible for corals to survive in the tropical ocean at such low temperatures. The assimilated mean tropical cooling by Tierney et al. (2019) is -3.9 °C (-4.2 °C to -3.7 °C, 95 % CI), which is larger than the data-only estimates that may suffer from incomplete and uneven sampling.

Regarding faunal assemblages, Ravelo et al. (1990) demonstrate that in the equatorial Atlantic they do not respond primarily to SST, but rather to thermocline and seasonality changes. Along similar lines, Telford et al. (2013) provide evidence that planktonic foraminifera assemblages can be more sensitive to subsurface temperatures than the 10-m SST that they are usually calibrated against, e.g., as in MARGO. They conclude that reconstructions of 10-m SST are likely to be biased, with the sign and magnitude of the bias varying regionally, probably causing a warm bias in the tropical North Atlantic, but foraminifera-based reconstructions for other ocean basins still remaining to be assessed

For many studies, the question is to what degree they included the Pacific Ocean in assessing the tropical cooling, or the high latitudes in assessing the global cooling; SST changes in the Arctic Ocean and the Southern Ocean are very small where permanent sea ice prevails. In our revised manuscript we will take up the above discussion and comment on why our estimate of the global SST decrease based on the MARGO faunal assemblages may be at the low end of the currently accepted range.

*Also: how was ECS calculated? There must be a scaling assumption to translate to global mean surface temperature, and then there has to be estimation of the forcing as well (the denominator). Please describe this.*

We used the same relationship as MARGO Project Members (2009), which is shown in Fig. 6 by Schneider von Deimling
et al. (2006).

*My overall take of this paper is: It's really interesting to see the application of DIVA to paleoclimate information, and this could use some more discussion and exploration (perhaps comparison to optimal interpolation). However in terms of providing new scientific insights into the LGM, the paper is limited here by use of the MARGO faunal dataset, which ultimately shapes the results. No matter what the method used, the MARGO data, particularly the assemblage data, provide an estimate of glacial*
*cooling that is very small. This result has been challenged a lot over the years and there is a sense that perhaps no-analog problems still plague the faunal data.*

*I think the best solution here would be to update the underlying dataset with new studies - either new faunal data or new faunal data + geochemical data. Otherwise, the conclusions of the paper re: glacial cooling and climate sensitivity are just the same as MARGO.*

*Alternatively, the authors could treat this paper as a methods paper. If the goal is to just demonstrate application of DIVA, then it's OK to stick with MARGO. But in that case comparisons should be made with other field estimation methods (OI, data as- similation) and the scientific results (LGM cooling and ECS) should be downplayed and presented critically since they are ultimately tied to the underlying data.*

We agree to elaborate more on the DIVA method and include a test of this method on a subset of the data. Updating the
MARGO data with new data sets is unfortunately beyond the scope of this manuscript, however, we will make clear that truly new results may not be expected and put the values of the LGM cooling and ECS, now based on a properly area- and uncertainty-weighted gridded climatology, in perspective, comparing them to values obtained from other data and different methods.

**Specific comments:**

*Abstract: Clarify that GLOMAP is based on only faunal transfer function data (except for the use of IP25 for sea ice).*

Agreed

*Section 2.1: Please clarify here what each reconstruction is based on (transfer functions, IP25, etc).*

Agreed

*Section 2.2: Why did you only use the faunal data from the MARGO collection? Also MARGO is now 11 years old. I imagine*
*that more data have been published since then. Certainly for the geochemical proxies this is true. I think it's worth updating the data with newly published results.*

See our response the major point no. 1

*Section 2.3: This section could benefit from a little more explanation of how DIVA works since most readers will not be familiar with Troupin et al. (2012). In particular, it would be useful to describe how DIVA is distinct from pure interpolation* 160 *(no information about spatial relationships) vs. optimal interpolation (covariance structure is set). Is DIVA essentially isotropic away from the coastlines?*

Thank you for bringing up this point. We will add to the manuscript that DIVA indeed makes use of a covariance structure. In our application, it is essentially isotropic away from the coastlines, however, in principle, if current velocities are known, an advective constraint may be imposed, too.

*Section 3.1: The use of past tense here is a little confusing. Use present tense for describing the results.*

Agreed, we will use present tense when describing the contents of Fig. 2 and Figs. A1 to A8.

**References**

Amrhein, D. E., Wunsch, C., Marchal, O., and Forget, G.: A Global Glacial Ocean State Estimate Constrained by Upper-Ocean Temperature Proxies, Journal of Climate, 31, 8059–8079, https://doi.org/10.1175/JCLI-D-17-0769.1, 2018.

[revised manuscript text omitted]

---

## Author Response (AR1)

**A global climatology of the ocean surface during the Last Glacial Maximum mapped on a regular grid (GLOMAP) – Final response to referee comments**

André Paul[1], Stefan Mulitza[1], Ruediger Stein[1, 2], and Martin Werner[2]

[1] MARUM – Center for Marine Environmental Sciences and Department of Geosciences, University of Bremen, Bremen, Germany
[2] Alfred Wegener Institute, Helmholtz Centre for Polar and Marine Research (AWI), Bremerhaven, Germany

**Response to Anonymous Referee #1**

We are thankful to the referee for the helpful and constructive comments that will surely improve our manuscript.

**General comments**

*The authors state is that the Data-Interpolating Variational Analysis method is also capable of analyzing much sparser data.*
*In the paper there is no real comparison or assessment of other methods that would allow the author to make this statement.*
*So I would encourage the authors to add a few lines explaining their choice, maybe by adding some details on the methods that*
*could justify their choice for the present study would be relevant and comparing with the methods employed to create CLIMAP*
*and GLAMAP climatologies.*

- In response to the valid concern raised by both referees, in the revised manuscript (p. 4, p. 6) we provide a test of the method by, as proposed by the second referee, withholding a certain fraction of the data, make a fit to the remaining data using DIVA, then compare the fit to the withheld data. We adopted the procedure by Schäfer-Neth et al. (2005), which allowed us compare our results to variogram analysis and kriging and the Levitus objective analysis.

- Furthermore, we took the opportunity to explain our choice of using DIVA (p. 4 of the revised manuscript) – for example, DIVA takes the coastlines into account, since an underlying variational principle is solved only on a finite-element mesh that covers the sea. This prevents the exchange of information across boundaries such as land bridges, peninsulas or islands, which might produce artificial mixing between, for example, Pacific and Atlantic water masses across the Panama isthmus. In solving the variational principle, it not only takes into account the distance between analysis and data, but also imposes a smoothness constraint and, if desired, an advection constraint, and moreover, it provides an uncertainty estimate.

- Regarding the method employed to create the CLIMAP climatology: As described in detail by Broccoli and Marciniak (1996; see also Manabe and Broccoli, 2020), CLIMAP used a subjective analysis procedure (i.e. contouring by hand) to

yield the paleoisotherm maps (CLIMAP Project Members, 1976, 1981), which were then digitized on a regular grid (see p. 2 of the revised manuscript).

- With respect to the GLAMAP climatology, different methods were applied: Contouring of the paleotemperature maps was also by hand, and the isotherms were derived by means of visual triangulation from strictly linear interpolation between the SST reconstructions at the irregularly distributed neighbor sites (Sarnthein et al., 2003; Pflaumann et al., 2003). For gridding, either the digitized isotherms (Paul and Schäfer-Neth, 2003) or the SST reconstructions at the sediment core positions (Schäfer-Neth and Paul, 2004) were objectively interpolated using variogram analysis and kriging in spherical coordinates; and the resulting gridded fields were compared (Schäfer-Neth and Paul, 2004, Fig. 5). The seasonal cycle was constructed in the same way as for the GLOMAP climatology: Following the PMIP (1993) guidelines, a sinusoidal cycle was fitted to the glacial-to-modern anomalies and then the modern monthly SST (taken as 10 m data from the WOA, 1998) was added.

- The variogram analysis and kriging cannot deal easily with coastlines, for example, it may take into account data points separated by a land bridge or an island. This was one motivation to apply the DIVA method (Troupin et al., 2012), which employs a finite-element mesh derived from a given topography.

*Data availability: it would be useful for the reader to have the direct URL to avoid searching within the PANGEA database. I searched using "GLOMAP" as keyword (https://www.pangaea.de/?q=GLOMAP) but that request did not return any result, so I guess the data will be published once the paper is published. What is the format of the climatology?*

Indeed, the final version of the data will be published in the PANGEA database once the paper is published. We already submitted our data to the PANGEA database. The format of the climatology is the Network Common Data Format (netCDF).

*Some parts of the processing workflow (Section 2) were not totally clear to me, for example lines 89-109: why are the two steps necessary, and why not use all the data at the same time for the monthly interpolations?*

We took two steps in order to make use of the diatom and radiolarian data from the Southern Ocean. Because in this region the biogenic particle flux to the sea floor is restricted to austral summer, even in areas unaffected by sea-ice cover (Abelmann and Gersonde, 1991; Gersonde and Zielinski, 2000; Fischer et al., 2002), only Southern Hemisphere summer (JFM) SST has been estimated by Gersonde et al. (2005). Therefore, in the first step, we only used the foraminiferal and dinoflagellate data for JAS and JFM. In the second step, we included the diatom and radiolarian data available for JFM and filled in the missing data for JAS by taking the results from the first step at the grid points where diatom and radiolaria data for JFM exist. In this way we were able to create monthly data at all grid points where data exist and repeat the DIVA analysis.

*lines 94-94: To each sea-ice covered data point we assigned an error of $2°C \longrightarrow$ does this mean that measurements taken where it is supposed to be sea ice are used for the gridding? From line 80, it seems that the finite-element mesh is based on a coastline from a glacial topography, so the measurements on ice would not influence the analysis.*

Indeed, we used a glacial topography to generate the coastlines, however, these coastlines do not reflect the sea-ice edges (lines 104-105 of the revised manuscript). Therefore the finite-element mesh extended to the sea-ice covered regions. To utilze the information on past sea-ice coverage, we digitized the sea-ice edges, created monthly sea-ice masks and included the seaice covered data points in the DIVA analysis. A relatively large error of 2 °C was assigned to each sea-ice covered data point to reflect the uncertainty in the LGM sea-ice extent reconstructions.

*Could you add the figure of the coastline and the finite-element mesh in the Appendix?*

We added maps of the coastlines and the finite-element meshes for both, the World Ocean Atlas (WOA) test of the DIVA method and the GLOMAP paleo-data analysis (Fig. A1).

**Minor comments and typos**

*39: This method allows to take $\longrightarrow$ allows one to take*

Corrected (line 52 of the revised manuscript)

*39: the uncertainty on the reconstruction $\longrightarrow$ the authors probably means the uncertainties on the observations (instrumental errors, representativeness errors etc).*

Indeed, we mean the paleo-data. Because these are not direct observations as instrumental data, but indirect estimates derived from fossil faunal assemblages using a transfer function technique, we often use the term "reconstruction", which may be confusing. Hence in the sentence in question we replaced "reconstruction" by "paleo-data" (line 53 of the revised manuscript).

*40: or for assessing the data-analysis mismatch. $\longrightarrow$ independently of the interpolation technique, the data-analysis mismatch is not always a relevant metrics: one can obtain a very small mismatch by forcing the analysis to be close to the observation. This would result in a "noisy" or "patchy" interpolated field, which may not represent what a climatology should be.*

This is certainly true, thank you for pointing this out. Figures 3 and 4 show that the interpolated fields are neither "noisy" or "patchy", something that we now mention in the discussion of our results (lines 246–248 of the revised manuscript).

*line 49: "we digitized sea-ice edges" $\longrightarrow$ can you explicit what is the process to digitize? For Xiao et al. (2015, Fig. 7a), the panel a of their figure did not display any coordinates, how was that solved? Does this also means that no other publication provides the sea-ice edges in digital format?*

It is indeed true that none of the publications provides the sea-ice edges in digital format. Therefore we had to digitize the curves from the published maps to obtain their location in geographic coordinates. In case of Xiao et al. (2015, Fig. 7a), neither the projection nor the coordinates are given, hence we used the few indicated topographic features (islands) and the sediment core locations to take into account a summer ice edge north of the Barents Sea in our sea-ice mask. We mention this in Section 2.1 of the revised manuscript.

*65 is associate with $\longrightarrow$ associated with*

Corrected

*72-73 the magnitude of the data (anomalies) themselves as well as on the gradients, the variability and data-analysis misfits*

Rewritten as follows: "the magnitude of the data (anomalies) themselves as well as on the gradients, the variability and data-analysis misfits"

*82 We fitted the covariance function to the foraminiferal data $\longrightarrow$ indicate how many data points were considered for the fit.*

There were 444 data points considered for the fit. We added this number to the revised manuscript.

90    *84 estimates of the correlation length of 9.2° and 10.2° ⟶ is there a physical explanation to this difference, or do you attribute that to numerics?*

We now attributed this difference to the overall larger data covariance for July-August-September (JAS), resulting in a slightly larger correlation length than for January-February-March (JFM).

*94 To each sea-ice covered data point we assigned an error of 2 °C ⟶ is it necessary, since you defined a coastline and*
95    *mask using "glacial topography GLAC-1D"*

Please see our response to your general comment on lines 94–94.

*103: new (artificial) diatom and radiolaria data ⟶ what is the source of these data? (and why "artificial"?)*

The diatom and radiolarian data for JAS are "artificial" in the sense that they were generated from the results of the first step of the DIVA analysis at the grid points where diatom and radiolaria data for JFM only exist (please see our response to your
100    general comment on lines 89–109).

*141 South Alantic ⟶ South Atlantic*

Corrected

*169-170 DIVA may be used to more accurately estimate the spatial covariances as described by the non-diagonal terms ⟶ Beckers et al. (2014) may be relevant for this aspect Beckers, J.-M.; Barth, A.; Troupin, C. & Alvera-Azcárate, A.*
105    *Some approximate and efficient methods to assess error fields in spatial gridding with DIVA (Data Interpolating Variational Analysis) (2014). Journal of Atmospheric and Oceanic Technology, 31: 515-530. doi:10.1175/JTECH-D-13-00130.1*

Thank you for pointing out this reference to us. We included it in the revised manuscript.

*224 may allow to first smooth ⟶ may allow one to first smooth*

Corrected

110    *Figure 2:*

– *indicate the meaning of the yellow-brownish area close to the Antarctica (rectangles in the attached figure)*

We apologize for the missing information in the figure caption. The yellow-brownish areas close to Antarctica and in the Arctic ocean indicate the LGM sea-ice masks based on the selected LGM sea-ice reconstructions.

– *Analyzed SST anomalies ⟶ with respect to what reference or background are computed the anomalies? (also in text,*
115    *line 97).*

The LGM estimates from the MARGO database are themselves anomalies that are calculated with respect to World Ocean Atlas 1998. We will clarify this in the Methods section of the revised manuscript.

– *the size of the dots representing the data is a little bit to large, so several dots overlap, especially in the northern part of the domain. In Figure 4 the dots are smaller.*

120    Thank you for pointing this out to us. In the final version of the manuscript, we will reduce the size of the dots representing the data and use the same size in all figures.

*334 WOA: World Ocean Atlas 1998 ⟶ why not use a more recent version of the World Ocean Atlas?*

The main reason for using the 1998 version of the World Ocean Atlas is to be consistent with the MARGO database, because it is this version that was used for calibrating the methods which were used to reconstruct the LGM estimates and served as a reference for the LGM SST anomalies. Furthermore, the majority of the data that entered the 1998 version represents observations that were taken before the intensification of global warming towards the end of the 20[th] century, thereby reducing the anthropogenic imprint.

*Figures for the monthly fields: having 6 (or maybe 12) sub-figures seems possible and won't cause problem to the readability of the plots.*

Thank you fo this suggestion. In the final version of the manuscript, we will try combining 6 or even 12 sub-plots in one figure and check its readability, especially of the dots representing the data.

*like IP25, but the SST reconstruction is based only on faunal data from the MARGO collection. Why is this? I'm not sure why*
*the authors would not use the geochemical data in the MARGO collection (?) If there is a reason, then it should be made clear.*
10  *There is arguably value in a single-proxy field reconstruction, but it should be justified. Also, I think some of the faunal data*
*may have no-analog issues. Were these dealt with in any way?*

We agree that we should make clear why we base the gridded SST on the faunal and floral data only, which we now summarize in the revised discussion (p. 10–11 of the revised manuscript). In fact, there are a number of reasons:

 – Our interest is in a monthly climatology of the SST anomaly and sea-ice extent during the LGM, and while faunal and
15      loral assemblages are not without issues, they are the only sedimentary proxy that can provide a seasonal reconstruction.

 – In addition, a single-proxy reconstruction has the advantage of internal consistency between the different sediment core
    sites, thereby reducing the noise.

 – As for the MARGO reconstruction of the northern North Atlantic Ocean, all four proxies (planktonic foraminifer as-
    semblages, dinocyst assemblages, alkenone coccolithophorid biomarkers and Mg/Ca ratios in planktonic forminifers)
20      support the same features of sea-ice cover (de Vernal et al., 2006; see also Sarnthein et al., 2003b). The IP25/PIP25

data by Xiao et al. (2015, Fig. 7a) and Méheust et al. (2018) add information for the Barents Sea and the North Pacific Ocean, respectively. However, in the Nordic Seas, there are large discrepancies between the different SST reconstructions, well above their level of uncertainties (de Vernal et al., 2006). We suspect that the apparently warm signal recorded by dinocyst assemblages, coccolithophores and alkenones may be due to long-distance lateral transport (de Vernal et al., 2006; Rühlemann and Butzin, 2006), whereas foraminifera-based proxies have the advantage that they drop very quickly to the sediment (Takahashi and Be, 1984). Further possible sources of errors are the overwintering of dinoflagellates in a cyst phase as well as their broad tolerances for temperature (Dale, 2001). Using alkenones for SST reconstructions might be problematic in high latitudes because of the low sensitivity of the calibration of alkenones at low temperature (Conte et al., 2006), the possibility of redeposition of old and warm signal carrying alkenones with particulate matter originating from the glaciated continental margins and once more the influence of alkenones transported by currents from warmer areas into the polar regions (e.g., Bendle and Rosell-Melé, 2004; Filippova et al., 2016).

– Furthermore, foraminiferal assemblages are usually dominated by species adapted to the environment of the overlying water column (Morey et al., 2005). We therefore consider the temperature estimation to be more robust against the expatriation of single shells that can affect proxies measured on monospecific samples.

– Finally, proxies based on the chemistry of shells of living organisms suffer from the inherent problem that the environmental sensitivity of that organism biases the recording of the proxy (Mix, 1987; Fraile et al., 2009). The transfer function method does not have this problem since it actually uses the environmental sensitivity of foraminifera.

The MARGO project carefully dealt with the no-analog problem in a number of ways. For example, Gersonde et al. (2005) discard all samples with no analogs (dissimilarity > 0.25) and when the majority of the samples in the LGM interval has no anlogs, the estimated quality level is downgraded to 3. Kucera et al. (2005) combine three methods (Artificial Neural Networks – ANN, Revised Analog Method – RAM, Maximum Similarity Technique – SIMMAX) in a multi-technique approach that facilitates a test of the robustness of SST estimates and provides a means to identify potential no-analog conditions or faunas.

*A bigger problem with the choice of data is that MARGO is over a decade old now, and surely there have been more faunal datasets published since then (Certainly, there is far more geochemical data available now). Since the authors just use the MARGO data, they get results that nearly the same as MARGO. This doesn't seem like an advance in our understanding of the LGM. If the purpose of this paper is provide new insights into the LGM, I would suggest that the authors consider updating their dataset. If the purpose of this paper is to demonstrate a method (DIVA) then the cooling and ECS results should be downplayed.*

An important purpose of our manuscript is indeed to demonstrate the applicability of DIVA to sparse and irregularly spaced paleoceanographic data. As we now point out in the Outlook section of the revised manuscript (p. 14), an update of MARGO is certainly important, however, it is beyond the scope of our current work.

We agree that we cannot expect to find truly new results, but we actually see a number of advantages using the well-established MARGO database: To date, it still provides the most comprehensive dataset of LGM SST anomalies at the sediment core locations including their error estimates. Without creating a spatially complete, gridded climatology the MARGO project

has not been able to provide area- and uncertainty-weighted global and regional averages or boundary conditions. The main motivation of our manuscript is to provide a spatially complete, continuous gridded climatology for such purposes. By using MARGO, we can directly compare the gridded and non-gridded data and assess the effect of proper weighting in calculating global and regional averages, and we indeed find small but noticeable differences.

Another advantage of the MARGO database is the consistent use of the modern SST (WOA, 1998) as the calibration as well as the reference dataset. Most methods used for temperature reconstruction are calibrated to absolute temperatures, not temperature differences or changes. By using the calibration dataset as a reference dataset, any error is only incurred once in calculating the LGM anomaly. If using any other time slice (e.g., the Late Holocene) as a reference dataset, the error would double.

With respect to downplaying the cooling and ECS results, please see our response to the major point no. 4.

*2) DIVA method. This method is new to me, but seems appropriate for the problem at hand. However some more description of the method is needed here for non-specialists. I'm also wondering, given that DIVA was designed to work with more dense modern oceanographic data, how well it does with the sparse data of the LGM? Can the authors do some validation tests to assess this? E.g., withhold 10-25% of the data, fit the field using DIVA, then validate on the withheld data? This would provide some sense of performance.*

Thank you for suggesting this useful test, which we now provide in the revised manuscript (p. 6 and p. 8) to give an idea of the performance, together with some more explanation of the method for non-specialists. For the test, we adopted the procedure by Schäfer-Neth et al. (2005), which allowed us compare our results to variogram analysis and kriging and the Levitus objective analysis.

*3) Comparisons to other field reconstructions of the LGM. The authors discuss how their result is fairly similar to MARGO, which is not surprising since the underlying data are similar. What about other products? There are some data assimilation prod- ucts to compare with (Annan and Hargreaves, 2013) - Paul is a co-author on one of them (Kurahaski-Nakamura et al., 2017 https://doi.org/10.1002/2016PA003001) and see also Amrhein et al., 2018 (https://doi.org/10.1175/JCLI-D-17-0769.1). We have a new data assimilation product available as well (in review, but a preprint is available here: https://eartharxiv.org/me5uj/) based on an updated database of geochemical proxies.*

Thank you for this valuable suggestion, too. In the revised manuscript, we follow up on it and compare our gridded clima-tology with the products by Annan and Hargreaves (2013), Kurahashi-Nakamura et al. (2017b) and Tierney et al. (2020) as well as by CLIMAP (1981) and GLAMAP (Sarnthein et al., 2003a) (Tables 3 and 4 and Figure 6), and we mention the advan-tages and disadvantages of the various methods. For example, while DIVA is a purely statistical method, the adjoint method used by Kurahashi-Nakamura et al. (2017a) (as well as Amrhein et al. (2018)) makes use of the physics and paramaterizations of an ocean general circulation model (MITgcm), however, it also inherits the biases of this model. In contrast, Annan and Hargreaves (2013) employ an ensemble of models, but with partly inconsistent physics and parameterizations.

*4) Estimates of glacial cooling and climate sensitivity. In keeping with the MARGO results, these are arguably unrealistically low (global SST change of -1.7, ECS of 1.5). The MARGO-based results of ECS (Schmittner et al. 2011) have faced a lot of criticism. Multiple studies have suggested a global SST change closer to -3C (Ballantyne et al., 2005, Lea et al., 2000) and a*

90  *corresponding global air temperature change closer to 5-6C (e.g. Snyder, 2016, Nature; Schneider von Deimling et al., 2006 GRL, Holden et al., 2010 Climate Dynamics, Bereiter et al., 2018, Nature, and our new estimate in our preprint noted above). There needs to be a critical discussion in light of these other results.*

According to Bindoff et al. (2013), the equilibrium climate sensitivity is likely in the range of 1.5 °C to 4.5 °C ("high confidence"), extremely unlikely less than 1°C ("high confidence"), and very unlikely greater than 6°C ("medium confidence").

95  The PALAEOSENS Project Members (2012) estimate based on a comprehensive set of reconstructions of past temperatures and radiative forcing yields a range of 2.2 °C to 4.8°C for the actual climate sensitivity. If taken at face value, our result would imply a value of the equilibrium climate sensitivity at the low end of the classical range. While neither a value of the equilibrium climate sensitivity at the low end (about 1.5 °C) nor at the high end (about 4.5 °C) can be ruled out beforehand, we agree that a critical discussion of our estimate, in the light of these results, and especially the new results by Judd et al. (2020), Sherwood

100  et al. (2020) and Tierney et al. (2020), is needed. In an attempt to provide a balanced view and avoid a simplistic approach, we extended and largely rewrote the Discussion (p. 10–13) and Conclusions (p. 13–14) sections in the revised manuscript.

Regarding the study by Schmittner et al. (2011), it was mainly criticized because of some methodological issues (e.g., missing atmospheric feedbacks, a definition of climate sensitivity that includes feedbacks associated with vegetation, an insufficiently steep lapse rate, underprediction of land temperatures and misfit of ocean data in mid- and high latitudes – see the

105  discussion in RealClimate, 2011). These resulted in a global cooling during the LGM in the range of 1.7 °C to 3.7 °C, which appears to be too small and possibly led to an underestimate of the climate sensitivity with a surprisingly low uncertainty range of 1.4 °C to 2.8 °C. Indeed, Annan and Hargreaves (2013), who also used MARGO as ocean data, arrived at a larger global cooling during the LGM of 3.1 °C to 4.7 °C, supposedly implying a larger climate sensitivity.

The best resolved alkenone-based SST estimates from the central Pacific show an SST change between 1.2 °C and 2 °C (Prahl

110  et al., 1989; Lee et al., 2001; de Garidel-Thoron et al., 2007). From a number of studies using Mg/Ca as well as alkenones, Lea (2004) finds a tropical cooling at the LGM by 2.8 °C ± 0.7 °C. Leduc et al. (2017) summarize the results of the Sensitivity of the Tropics (SENSETROP) working group, which after the incorporation of high-quality records and a thorough quality control obtains a cooling of the low latitudes during the LGM of -2.3 °C ± 0.8 °C and -2.4 °C ± 0.8°C for alkenone- and Mg/Ca-based SST estimates, respectively. Tierney et al. (2020) obtain a very similar mean tropical cooling of -2.5 °C (-2.8 °C to -2.2 °C, 95

115  % CI) from the SST proxies on their own. These values are indeed larger than the estimates by CLIMAP, MARGO and Annan and Hargreaves (2013) by up to a degree, but not as large as the early estimate from corals by Guilderson et al. (1994) of about 5 °C, which made Crowley (2000) wonder whether it would be possible for corals to survive in the tropical ocean at such low temperatures. The assimilated mean tropical cooling by Tierney et al. (2020) is -3.9 °C (-4.2 °C to -3.7 °C, 95 % CI), which is larger than the data-only estimates that may suffer from incomplete and uneven sampling.

120  Regarding faunal assemblages, Ravelo et al. (1990) demonstrate that in the equatorial Atlantic they do not respond primarily to SST, but rather to thermocline and seasonality changes. Along similar lines, Telford et al. (2013) provide evidence that planktonic foraminifera assemblages can be more sensitive to subsurface temperatures than the 10-m SST that they are usually calibrated against, e.g., as in MARGO. They conclude that reconstructions of 10-m SST are likely to be biased, with the sign

and magnitude of the bias varying regionally, probably causing a warm bias in the tropical North Atlantic, but foraminifera-based reconstructions for other ocean basins still remaining to be assessed

In our revised manuscript, we took up the above discussion and commented on why our estimate of the global SST decrease based on the MARGO faunal and floral assemblages may be at the low end of the currently accepted range (p. 11–12 of the revised manuscript).

*Also: how was ECS calculated? There must be a scaling assumption to translate to global mean surface temperature, and then there has to be estimation of the forcing as well (the denominator). Please describe this.*

We used the same simple linear relationship as MARGO Project Members (2009), which is shown in Fig. 6 by Schneider von Deimling et al. (2006), but we now refer to the excellent recent review by Sherwood et al. (2020), too (lines 374–389 of the revised manuscript).

*My overall take of this paper is: It's really interesting to see the application of DIVA to paleoclimate information, and this could use some more discussion and exploration (perhaps comparison to optimal interpolation). However in terms of providing new scientific insights into the LGM, the paper is limited here by use of the MARGO faunal dataset, which ultimately shapes the results. No matter what the method used, the MARGO data, particularly the assemblage data, provide an estimate of glacial cooling that is very small. This result has been challenged a lot over the years and there is a sense that perhaps no-analog problems still plague the faunal data.*

*I think the best solution here would be to update the underlying dataset with new studies - either new faunal data or new faunal data + geochemical data. Otherwise, the conclusions of the paper re: glacial cooling and climate sensitivity are just the same as MARGO.*

*Alternatively, the authors could treat this paper as a methods paper. If the goal is to just demonstrate application of DIVA, then it's OK to stick with MARGO. But in that case comparisons should be made with other field estimation methods (OI, data as-similation) and the scientific results (LGM cooling and ECS) should be downplayed and presented critically since they are ultimately tied to the underlying data.*

We agree to elaborate more on the DIVA method and included a test of this method on a subset of the data. Updating the MARGO data with new data sets is unfortunately beyond the scope of the current manuscript and must be deferred to a new project. We put the values of the LGM cooling and ECS, now based on a properly area- and uncertainty-weighted gridded climatology, in perspective, comparing them to values obtained from other data and different methods.

**Specific comments:**

*Abstract: Clarify that GLOMAP is based on only faunal transfer function data (except for the use of IP25 for sea ice).*
Agreed (line 3 of the revised manuscript)
*Section 2.1: Please clarify here what each reconstruction is based on (transfer functions, IP25, etc).*
Agreed (lines 59-63 and line 65 of the revised manuscript)

*Section 2.2: Why did you only use the faunal data from the MARGO collection? Also MARGO is now 11 years old. I imagine that more data have been published since then. Certainly for the geochemical proxies this is true. I think it's worth updating the data with newly published results.*

See our response the major point no. 1

*Section 2.3: This section could benefit from a little more explanation of how DIVA works since most readers will not be familiar with Troupin et al. (2012). In particular, it would be useful to describe how DIVA is distinct from pure interpolation (no information about spatial relationships) vs. optimal interpolation (covariance structure is set). Is DIVA essentially isotropic away from the coastlines?*

Thank you for bringing up this point. We added to the manuscript that DIVA indeed makes use of information on spatial relationships (lines 99-102 of the revised manuscript). In our application, it is essentially isotropic away from the coastlines, however, in principle, if current velocities are known, an advective constraint may be imposed, too.

*Section 3.1: The use of past tense here is a little confusing. Use present tense for describing the results.*

Generally, we would like to adhere to the rule of describing our own results in past tense and referring to other published results in present tense.

**References**

Amrhein, D. E., Wunsch, C., Marchal, O., and Forget, G.: A Global Glacial Ocean State Estimate Constrained by Upper-Ocean Temperature Proxies, Journal of Climate, 31, 8059–8079, https://doi.org/10.1175/JCLI-D-17-0769.1, 2018.

[revised manuscript text omitted]
 used the annual-mean temperature for 10 m depth from the unanalyzed World Ocean Atlas 1998 data (WOA, 1998), which had also been used to calibrate the MARGO transfer function technique. These temperature data had been averaged to a $2° \times 2°$ regular grid, but otherwise had not been subject to any analysis or interpolation method. The coverage of the unanalyzed data is nearly complete, except for the central Arctic and some points off the Antarctic coast. This allows for a near-global comparison with the results of the interpolation.

The input data set for the DIVA method was created by binning all MARGO core locations in $2° \times 2°$ squares and sampling the $2° \times 2°$ WOA data at these points to reflect the density of the MARGO data. The DIVA method was used to interpolate the sampled points back to the $2° \times 2°$ grid and the annual-mean SST 
[revised manuscript text omitted]

[Figure]

**Figure A1.** Coastline contours and finite-element meshes for "'original grids"'. Top: for the WOA test of the DIVA method (centered on $210^\circ$ W). Bottom: for the GLOMAP analysis (centered on $180^\circ$ W)

610 **Appendix A: Maps of coastlines and finite-element meshes, monthly SST anomalies and their estimated uncertainties**

[Figure]

**Figure A2.** Sea-surface temperature anomaly and sea-ice extent for January, February and March. For February, we also show the MARGO reconstruction at the sediment core locations.

[Figure]

**Figure A3.** Uncertainty of SST anomaly for January, February and March. For February, we also show the estimated error of the MARGO reconstruction at the sediment core locations.

[Figure]

**Figure A4.** Sea-surface temperature anomaly and sea-ice extent for April, May and June

[Figure]

**Figure A5.** Uncertainty of SST anomaly for April, May and June

[Figure]

**Figure A6.** Sea-surface temperature anomaly and sea-ice extent for July, August and September. For August, we also show the MARGO reconstruction at the sediment core locations.

[Figure]

**Figure A7.** Uncertainty of SST anomaly for July, August and September. For August, we also show the estimated error of the MARGO reconstruction at the sediment core locations.

[Figure]

**Figure A8.** Sea-surface temperature anomaly and sea-ice extent for October, November and December

[Figure]

**Figure A9.** Uncertainty of SST anomaly for October, November and December

---

## Referee Report (RR1)

The paper described GLOMAP, new climatology for sea-surface temperature and sea-ice extension during the Last Glacial Maximum. The climatology is created using the Data-Interpolating Variational Analysis (DIVA) software tool and based on sparse SST reconstructions.

**Scientific significance:** the manuscript represents a substantial contribution in the terms of data (i.e. a new climatology) and application of a method (DIVA) to a specific type of data, with a low spatial coverage. The final products (climatology) will certainly be useful to other scientists and employed in different contexts.

**Scientific quality:** the scientific approach and applied methods are valid: the data processing is well described, the limitations and the uncertainties on the data are clearly presented. The comparisons with other gridding techniques is particularly relevant.

**Presentation quality:** the manuscript and the figures are clear, the document is concise and well structured.

**General comments**

109-118 These 2 paragraphs are not totally clear to me: I understood that $2°$ by $2°$ averaged data are used, but then it is stated (line 117) that "DIVA method was used to interpolate the sampled points back to the 2 2 grid". I might miss something, but I would appreciate if you could make it clearer, maybe adding a figure.

247 when you indicate "Figures 3 and 5 show that when applied to the paleo data the interpolated fields are neither noisy nor patchy"

it would be relevant to be complete and indicate that this is true because of the selected analysis parameters. For instance, working with a very small correlation length ($L \approx 0.2°$ for instance) would have led to a noisy fields.

In DIVA, one can select different coordinate systems (Section 6.2.3 in the User Manual): could you indicate which one was used (probably icoord=2, cosine projection).

255 "but thanks to the underlying global finite-element mesh with less complications..." $\rightarrow$ is there a benefit in terms of computational time that could be mentioned here?

Figure 2. In the workflow, there is a final step not visible there: the analysis itself, performed after the estimation of the analysis parameters.

Figure 3: it would be interesting to also have the number of data points for each period.

Figure 6: it seems that the GLOMAP product is the only one properly dealing with the Mediterranean Sea: the spatial resolution in Annan and Hargreaves (2013) represents the Mediterranean as two different sub-seas; in Kurahashi-Nakamura et al. (2017) the Mediterranean is absent; in Tierney et al. (2020), it appears homogeneous. I guess this does not impact the result of the studies performed at a local scale, but it might be worth mentioning this difference when comparing the methods, especially if one takes into account that there are available data in the Mediterranean Sea (Figure 1 for example).

General: the term SST is employed frequently, yet it is often referring to the temperature at 10 meters. Is it correct? I believe that in remote-sensing and in operational oceanography, SST refers to the temperature in the first millimeters of the water column. Could you address that definition early in the manuscript?

**Minor comments and typos**

The sea-ice mask seems to be a time-demanding product itself, is it also made available for re-use?

30 (MARGO) (Kucera et al., 2005a). $\rightarrow$ (MARGO, Kucera et al., 2005a). using the `\citep[][]{}` command in latex

35 (see also Broccoli and Marciniak (1996 and Manabe and Broccoli, 2020). $\rightarrow$ missing parenthesis

110 We used the annual-mean temperature for 10 m depth $\rightarrow$ at 10 m depth

126 we fixed the correlation length at average value of $10°$ $\rightarrow$ at an average value of

153 SH $\rightarrow$ please define (Southern Hemisphere I guess)

167 from the modern topography $\rightarrow$ indicate which topography was employed (including the version number)

179 the impact of advection by the western boundary currents, which is missing in our application of DIVA $\rightarrow$ the other methods don't use the advection neither in the interpolation scheme, so the difference should not come

187 There was even an $1°C$ $\rightarrow$ There was even a $1°C$

227 Eq (5): what is the denominator $u_i^2$? Also, the sum should be written $\sum_{i=1}^{N_{data}}$

318 has no anlogs, $\rightarrow$ analogs

394 than assimilationg $\rightarrow$ than assimilation

---

## Author Response (AR2)

**A global climatology of the ocean surface during the Last Glacial Maximum mapped on a regular grid (GLOMAP) – Final response to referee comments**

André Paul[1], Stefan Mulitza[1], Ruediger Stein[1, 2], and Martin Werner[2]

[1] MARUM – Center for Marine Environmental Sciences and Department of Geosciences, University of Bremen, Bremen, Germany
[2] Alfred Wegener Institute, Helmholtz Centre for Polar and Marine Research (AWI), Bremerhaven, Germany

**Response to Anonymous Referee #1**

We are very grateful to the referee for sharing her/his expertise with us and bringing up several key points that strengthen our manuscript considerably.

**General comments**

5 *109-118 These 2 paragraphs are not totally clear to me: I understood that $2°$ by $2°$ averaged data are used, but then it is stated (line 117) that "DIVA method was used to interpolate the sampled points back to the $2° \times 2°$ grid". I might miss something, but I would appreciate if you could make it clearer, maybe adding a figure.*

Thank you for pointing this out to us. We admit that our original formulation was indeed not totally clear. To make it clearer, we extended the section in question and added a new figure that details the step of testing the DIVA method. In addition, we
10 extended the caption of Fig. 2 (previously Fig. 1):

«To first test the DIVA method on data that are much sparser than oceanographic observations, we adopted the procedure by Schäfer-Neth et al. (2005). We took the test data from the World Ocean Atlas 1998. According to WOA (1998), the original ocean profile data are first vertically interpolated from observed depth levels to standard depth levels, then the arithmetic means of each variable in each $1°$ and $5°$ square of the World Ocean are calculated. Except for calculating the arithmetic means, these
15 data have not been subject to any other analysis. These global fields are therefore referred to as "unanalyzed" fields. The $1°$ unanalyzed annual-mean temperature at a depth of 10 m had been used to calibrate the MARGO transfer function technique (the original data file name is `t00mn1`; it is also available as `otemp.raw1deg.nc` from psl.noaa.gov).

Schäfer-Neth et al. (2005) further binned these data into a regular grid with a constant resolution of $2°$ using the GMT program `xyz2grd` (Wessel and Smith, 1998). The coverage is nearly complete, except for the central Arctic Ocean and some
20 points off the Antarctic coast (cf. Fig. 1, *top*). Finally, they greatly reduced this coverage by keeping only those $2°$ squares that contain an ocean sediment core site from MARGO Project Members (2009). This is the input data set for testing the DIVA method (cf. Fig. 1, *middle*).

The DIVA method was used to interpolate these sparse test data to a complete regular grid with a constant resolution of 2°. The differences between the interpolated field (Fig. 1, *bottom*) and the "unanalyzed" field (Fig. 1, *top*) were calculated as a measure of the misfit. This allows for a near-global assessment of the result from the interpolation using the DIVA method.»

*247 when you indicate "Figures 3 and 5 show that when applied to the paleo data the interpolated fields are neither noisy nor patchy" it would be relevant to be complete and indicate that this is true because of the selected analysis parameters. For instance, working with a very small correlation length ($L \approx 0.2\circ$ for instance) would have led to a noisy fields.*

We rephrased that sentence as follows:

«Figures 4 and 6 show that when applied to the paleo data the interpolated fields are neither "noisy" nor "patchy". Because the paleo data allowed for a large correlation length of 10°, we obtained a smooth climatology, which we take as an indication that the data points were not overfitted.»

*In DIVA, one can select different coordinate systems (Section 6.2.3 in the User Manual): could you indicate which one was used (probably icoord=2, cosine projection).*

Yes, it is true that we used the parameter `icoordchange = 2` that corresponds to the cosine projection. We now mention this in Subsection 2.3 of the Methods section:

«To apply the DIVA method to the paleo data, we used the glacial topography GLAC-1D at 21,000 years before present (cf. Tarasov et al., 2012; Briggs et al., 2014) to generate glacial coast lines and create a corresponding global finite-element mesh using a cosine projection.»

*245 "but thanks to the underlying global finite-element mesh with less complications..." ⟶ is there a benefit in terms of computational time that could be mentioned here?*

In fact, it is difficult to directly compare the requirement on computing time, but our impression is that overall the method is more efficient and needs less time. Therefore we added a small note as follows:

«We indeed found that the DIVA method was capable of analyzing data that were much sparser than current oceanographic observations, with a skill that was comparable to variogram analysis and kriging, but thanks to the underlying global finite-element mesh with less complications (such as the introduction of communication masks to avoid the pairing of data points that are unlikely to influence each other in the real ocean, cf. Schäfer-Neth et al., 2005, Fig. 2) and overall in less time.»

*Figure 2. In the workflow, there is a final step not visible there: the analysis itself, performed after the estimation of the analysis parameters.*

This is certainly true, accordingly, we added the final step to the workflow to Fig. 3 (previously Fig. 2):

*Figure 3: it would be interesting to also have the number of data points for each period.*

The foraminiferal and dinoflagellate data for JAS and JFM are comprised of 464 data points. In addition, there are 117 points from the diatom data and 19 points from the radiolarian data for Southern Hemisphere summer (JFM). In total, there are 464 data points for JAS and 600 data points for JFM (without data points that are assumed to be covered by sea ice according to our sea-ice reconstruction).

We added this information to Subsection 2.3 of the Methods section as well as to the caption of Fig. 4 (previously Fig.3):

«Therefore, in the first step, we only used the foraminiferal and dinoflagellate data for JAS and JFM (464 data points). In the second step, we included the diatom data (117 data points) and radiolarian data (19 data points) available for JFM and filled in the missing data for JAS by taking the results from the first step at the grid points where diatom and radiolaria data for JFM exist.»

*Figure 6: it seems that the GLOMAP product is the only one properly dealing with the Mediterranean Sea: the spatial resolution in Annan and Hargreaves (2013) represents the Mediterranean as two different sub-seas; in Kurahashi-Nakamura et al. (2017) the Mediterranean is absent; in Tierney et al. (2020), it appears homogeneous. I guess this does not impact the result of the studies performed at a local scale, but it might be worth mentioning this difference when comparing the methods, especially if one takes into account that there are available data in the Mediterranean Sea (Figure 1 for example).*

To properly deal with this difference between the methods, we first rephrased Secion 2.4 of the Methods section:

«For a a comparison to other reconstructions, we selected the recent studies by Annan and Hargreaves (2013), Kurahashi-Nakamura et al. (2017) and Tierney et al. (2020) as well as the earlier studies by CLIMAP (1981) and GLAMAP (Sarnthein et al., 2003). The horizontal resolution differs among these reconstructions and ranges between $1°$ and $5°$. For analysis and plotting purposes, we interpolated them to the same regular grid with a constant resolution of $1°$. We calculated the annual-mean anomalies for the global, tropical and high-latitude oceans from these studies as well as our own results. Because an uncertainty estimate was not available for all studies, we only weighted by area.»

We added the following sentence to the caption of Table 3:

«The NSST results from the multiple linear regression by Annan and Hargreaves (2013) are provided on a regular grid with a constant resolution of $5°$.»

We added a new paragraph to the Discussion section:

«Regarding the Mediterranean Sea, in the coarse reconstruction by Annan and Hargreaves (2013) it is represented as two separated "sub-seas", while it is completely missing in the reconstruction by Kurahashi-Nakamura et al. (2017). The "off-line" data assimilation method by Tierney et al. (2020) yields a homogeneous result. It seems that the GLOMAP reconstruction is the only one that can properly present the Mediterranean Sea, in terms of spatial resolution of the underlying finite-element mesh as well as in taking into account the available data (cf. Fig. 4 and  Fig. 6)».

*General: the term SST is employed frequently, yet it is often referring to the temperature at 10 meters. Is it correct? I believe that in remote-sensing and in operational oceanography, SST refers to the temperature in the first millimeters of the water column. Could you address that definition early in the manuscript?*

In response to this valid remark, we now distinguish between sea-surface temperature (SST) and near-sea surface temperature (NSST) and added the following paragraph to the Introduction:

«Following, e.g., Dail and Wunsch (2014), the adjective "near" is used to distinguish these temperatures, which in the case of the GLAMAP and MARGO projects are based on calibrations for the top 10 m of the ocean and depend on phytoplankton and zooplankton that live even deeper, in the top 200 m to 300 m of the ocean, from those used in other communities, in which SST is at the surface itself and can even be a skin temperature that does not reflect the temperature below.»

**Minor comments and typos**

*The sea-ice mask seems to be a time-demanding product itself, is it also made available for re-use?*

Yes, we will make the sea-ice mask available along with the NSST reconstruction.

*30 (MARGO) (Kucera et al., 2005a). $\longrightarrow$ (MARGO, Kucera et al., 2005a). using the () command in latex*

*35 (see also Broccoli and Marciniak (1996 and Manabe and Broccoli, 2020). $\longrightarrow$ missing parenthesis*

*110 We used the annual-mean temperature for 10 m depth $\longrightarrow$ at 10 m depth*

*126 we fixed the correlation length at average value of 10∘ $\longrightarrow$ at an average value of*

All done

*153 SH $\longrightarrow$ please define (Southern Hemisphere I guess)*

Now we use "Southern Hemisphere" throughout instead.

*167 from the modern topography $\longrightarrow$ indicate which topography was employed (including the version number)*

We employed the bottom depth assigned to each 1° square of the World Ocean Atlas 2018 as topography, and we modified the paragraph (as well as the caption of Figure A1) accordingly:

«Figure A1 shows the coastlines that were generated from the modern topography (based on the bottom depth assigned to each 1° square by Garcia et al., 2019) for testing the DIVA method on the WOA (1998) data sampled at the MARGO core locations, as well as from the LGM topography (GLAC-1D, cf. Tarasov et al., 2012; Briggs et al., 2014) for our application of the DIVA method to the LGM NSST reconstruction.»

*179 the impact of advection by the western boundary currents, which is missing in our application of DIVA $\longrightarrow$ the other methods don't use the advection neither in the interpolation scheme, so the difference should not come*

This is correct, hence we dropped the last half-sentence from the manuscript.

*187 There was even an 1 °C $\longrightarrow$ There was even a 1 °C*

Done

*227 Eq (5): what is the denominator $u^2$? Also, the sum should be written $\sum^{N_{\text{data}}}$*

We rewrote Eq. 5 as follows:

$$J_{\text{misfit}} = \frac{1}{N_{\text{data}}} \sum_{i=1}^{N_{\text{data}}} \frac{\left(T_i^{\text{gridded}} - T_i^{\text{data}}\right)^2}{e_i^2} \ . \tag{1}$$

In this equation, $e_i$ is the average uncertainty of the data in the $i$th grid cell. We now use the symbol $e_i$ to distinguish it from $u_i$, which is the uncertainty of the analyzed field in the $i$th grid cell.

*318 has no anlogs, $\longrightarrow$ analogs*

Done

*394 than assimilationg $\longrightarrow$ than assimilation*

Done

**References**

[revised manuscript text omitted]